# GOAL-SPACE PLANNING WITH SUBGOAL MODELS

## ABSTRACT

This paper investigates a new approach to model-based reinforcement learning using background planning: mixing (approximate) dynamic programming updates and model-free updates, similar to the Dyna architecture. Background planning with learned models is often worse than model-free alternatives, such as Double DQN, even though the former uses significantly more memory and computation. The fundamental problem is that learned models can be inaccurate and often generate invalid states, especially when iterated many steps. In this paper, we avoid this limitation by constraining background planning to a set of (abstract) subgoals and learning only local, subgoal-conditioned models. This goal-space planning (GSP) approach is more computationally efficient, naturally incorporates temporal abstraction for faster long-horizon planning and avoids learning the transition dynamics entirely. We show that our GSP algorithm can learn significantly faster than a Double DQN baseline in a variety of situations.

## 1 INTRODUCTION

Planning with learned models in reinforcement learning (RL) is important for sample efficiency. Planning provides a mechanism for the agent to simulate data, in the background during interaction, to improve value estimates. Dyna (Sutton, 1990) is a classic example of background planning. On each step, the agent simulates several transitions according to its model, and updates with those transitions as if they were real experience. Learning and using such a model is worthwhile in vast or ever-changing environments, where the agent learns over a long time period and can benefit from re-using knowledge about the environment.

The promise of Dyna is that we can exploit the Markov structure in the RL formalism, to learn and adapt value estimates efficiently, but many open problems remain to make it more widely useful. These include that 1) one-step models learned in Dyna can be difficult to use for long-horizon planning, 2) learning probabilities over outcome states can be complex, especially for high-dimensional states and 3) planning itself can be computationally expensive for large state spaces.

A variety of strategies have been proposed to improve long-horizon planning. Incorporating options as additional (macro) actions in planning is one approach. An *option* is a policy coupled with a termination condition and initiation set (Sutton et al., 1999). They provide temporally-extended ways of behaving, allowing the agent to reason about outcomes further into the future. Incorporating options into planning is a central motivation of this paper, particularly how to do so under function approximation. Options for planning has largely only been tested in tabular settings (Sutton et al., 1999; Singh et al., 2004; Wan et al., 2021). Recent work has considered mechanism for identifying and learning option policies for planning under function approximation (Sutton et al., 2022), but as yet did not consider issues with learning the models.

A variety of other approaches have been developed to handle issues with learning and iterating one-step models. Several papers have shown that using forward model simulations can produce simulated states that result in catastrophically misleading values (Jafferjee et al., 2020; van Hasselt et al., 2019; Lambert et al., 2022). This problem has been tackled by using reverse models (Pan et al., 2018; Jafferjee et al., 2020; van Hasselt et al., 2019); primarily using the model for decision-time planning (van Hasselt et al., 2019; Silver et al., 2008; Chelu et al., 2020); and improving training strategies to account for accumulated errors in rollouts (Talvitie, 2014; Venkatraman et al., 2015; Talvitie, 2017). An emerging trend is to avoid approximating the true transition dynamics, and instead learn dynamics tailored to predicting values on the next step correctly (Farahmand et al., 2017;

Farahmand, 2018; Ayoub et al., 2020). This trend is also implicit in the variety of techniques that encode the planning procedure into neural network architectures that can then be trained end-to-end (Tamar et al., 2016; Silver et al., 2017; Oh et al., 2017; Weber et al., 2017; Farquhar et al., 2018; Schrittwieser et al., 2020). We similarly attempt to avoid issues with iterating models, but do so by considering a different type of model.

Much less work has been done for the third problem in Dyna: the expense of planning. There is, however, a large literature on approximate dynamic programming—where the model is given—that is focused on efficient planning (see (Powell, 2009)). Particularly relevant to this work is restricting value iteration to a small subset of landmark states (Mann et al., 2015). The resulting policy is suboptimal, restricted to going between landmark states, but planning is provably more efficient.

The use of landmark states has also been explored in *goal-conditioned RL*, where the agent is given a desired goal state or states. The first work to exploit the idea of landmark states was for learning and using universal value function approximators (UVFAs) (Huang et al., 2019). The UVFA conditions action-values on both state-action pairs as well as landmark states. The agent can reach new goals by searching on a learned graph between landmark states, to identify which landmark to moves towards. A flurry of work followed, still in the goal-conditioned setting (Nasiriany et al., 2019; Emmons et al., 2020; Zhang et al., 2020; 2021; Aubret et al., 2021; Hoang et al., 2021; Gieselmann & Pokorny, 2021; Kim et al., 2021; Dubey et al., 2021).

In this paper, we exploit the idea behind landmark states for efficient background planning in general online reinforcement learning problems. The key novelty is a framework to use *subgoal-conditioned models*: temporally-extended models that condition on subgoals. The models are designed to be simpler to learn, as they are only learned for states local to subgoals and they avoid generating entire next state vectors. We use background planning on subgoals, to quickly propagate (suboptimal) value estimates for subgoals. We propose subgoal-value bootstrapping, that leverages these quickly computed subgoal values, but mitigates suboptimality by incorporating an update on real experience. We prove that dynamic programming with our subgoal models is sound (Proposition 2) and that our modified update converges, and in fact converges faster due to reducing the effective horizon (Proposition 3). We show in the PinBall environment that our Goal-Space Planning (GSP) algorithm can learn significantly faster than Double DQN, and still reaches nearly the same level of performance.

## 2 PROBLEM FORMULATION

We consider the standard reinforcement learning setting, where an agent learns to make decisions through interaction with an environment, formulated as Markov Decision Process (MDP) $(\mathcal{S}, \mathcal{A}, \mathcal{R}, \mathcal{P})$. $\mathcal{S}$ is the state space and $\mathcal{A}$ the action space. $\mathcal{R} : \mathcal{S} \times \mathcal{A} \times \mathcal{S} \to \mathbb{R}$ and the transition probability $\mathcal{P} : \mathcal{S} \times \mathcal{A} \times \mathcal{S} \to [0, \infty)$ describes the expected reward and probability of transitioning to a state, for a given state and action. On each discrete timestep $t$ the agent selects an action $A_t$ in state $S_t$, the environment transitions to a new state $S_{t+1}$ and emits a scalar reward $R_{t+1}$.

The agent's objective is to find a policy $\pi : S \times A \to [0, 1]$ that maximizes expected *return*, the future discounted reward $G_t \doteq R_{t+1} + \gamma_{t+1} G_{t+1}$. The state-based discount $\gamma_{t+1} \in [0, 1]$ depends on $S_{t+1}$ (Sutton et al., 2011), which allows us to specify termination. If $S_{t+1}$ is a terminal state, then $\gamma_{t+1} = 0$; else, $\gamma_{t+1} = \gamma_c$ for some constant $\gamma_c \in [0, 1]$. The policy can be learned using algorithms like Q-learning (Sutton & Barto, 2018), which approximate the action-values: the expected return from a given state and action.

We can incorporate models and planning to improve sample efficiency beyond these basic model-free algorithms. In this work, we focus on background planning algorithms: those that learn a model during online interaction and asynchronously update value estimates use dynamic programming updates. The classic example of background planning is Dyna (Sutton, 1990), which performs planning steps by selecting previously observed states, generating transitions—outcome rewards and next states—for every action and performing a Q-learning update with those simulated transitions.

Planning with learned models, however, has several issues. First, even with perfect models, it can be computationally expensive. Running dynamic programming can require multiple sweeps, which is infeasible over a large number of states. A small number of updates, on the other hand, may be insufficient. Computation can be focused by carefully selecting which states to sample transitions

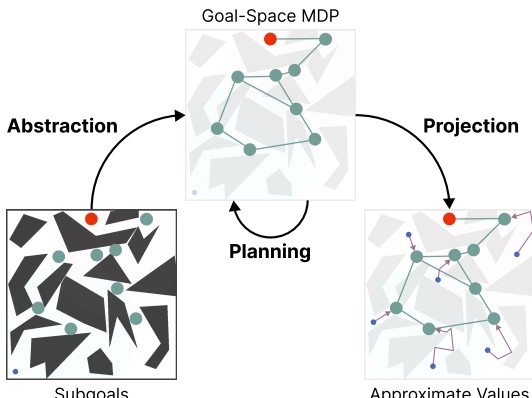

Goal-Space MDP

Abstraction

Projection

Planning

Subgoals

Approximate Values

Figure 1: Goal-Space Planning in the Pinball environment (see Section 4.1). The agent begins with a set of subgoals (denoted in teal) and learns a set of subgoal-conditioned models. (**Abstraction**) Using these models, the agent forms an abstract goal-space MDP where the states are subgoals with options to reach each subgoal as actions. (**Planning**) The agent plans in this abstract MDP to quickly learn the values of these subgoals. (**Projection**) Using learned subgoal values, the agent obtains approximate values of states based on nearby subgoals and their values. These quickly updated approximate values are then used to speed up learning.

from—called search control—but how to do so effectively remains largely unanswered with only a handful of works (Moore & Atkeson, 1993; Wingate et al., 2005; Pan et al., 2019).

The second difficulty arises due to errors in the learned models. In reinforcement learning, the transition dynamics is represented with an expectation model $\mathbb{E}[S'|s, a]$ or a probabilistic model $P(s'|s, a)$. If the state space or feature space is large, then the expected next state or distribution over it can be difficult to estimate, as has been repeatedly shown (Talvitie, 2017). Further, these errors can compound when iterating the model forward or backward (Jafferjee et al., 2020; van Hasselt et al., 2019). It is common to use an expectation model, but unless the environment is deterministic or we are only learning the values rather than action-values, this model can result in invalid states and detrimental updates (Wan et al., 2019).

In this work, we take steps towards the ambitious question: how can we leverage a separate computational procedure (planning with a model) to improve learning in complex environments? More specifically, we consider background planning for value-based methods. We address the two difficulties with classic background planning strategies discussed above, by focusing planning on a set of subgoals (abstract states) and changing the form of the model.

## 3 GOAL-SPACE PLANNING WITH SUBGOAL-CONDITIONED MODELS

At a high level, the Goal-Space Planning algorithm focuses planning over a set of given abstract subgoals to provide quickly updated approximate values to speed up learning. To do so, the agent first learns a set of *subgoal-conditioned models*, minimal models focused around planning utility. These models then forms a temporally abstract goal-space MDP, with subgoals as states, and options to achieve each subgoal as actions. Finally, the agent can update its policy based on these subgoal values to speed up learning. Figure 1 provides a visual overview of this process.

### 3.1 DEFINING SUBGOALS

Assume we have a finite subset of subgoal vectors $\mathcal{G}$. For example, $g$ could correspond to a situation where both the front and side distance sensors of a robot report low readings—what a person would call being in a corner. This $g$ could be represented using a two-dimensional vector, even if the sensory space is 100-dimensional. In general, subgoals need not be instances of states (i.e., $\mathcal{G} \not\subset \mathcal{S}$). As another example, in Figure 1, we simply encode the nine subgoals—which correspond to regions with a small radius—using a tabular encoding of nine one-hot vectors. Essentially, our subgoals define a new state space in an abstract MDP, and these new abstract states (subgoals) can be encoded or represented in different ways, just like in regular MDPs.

To fully specify a subgoal, we need a *membership function* $m$ that indicates if a state $s$ is a member of subgoal $g$: $m(s, g) = 1$, and zero otherwise. Many states can be mapped to the same subgoal $g$. For the above example, if the first two elements of the state vector $s$ consist of the front and side distance sensor, $m(s, g) = 1$ for any states where $s_1, s_2$ are less than some threshold $\epsilon$. For a concrete example, we visualize subgoals for the environment in our experiments in Figure 1.

Finally, we only reason about reaching subgoals from a subset of states, called *initiation sets* for options (Sutton et al., 1999). This constraint is key for locality, to learn and reason about a subset of states for a subgoal. We assume the existence of a (learned) *initiation function* $d(s, g)$ that is 1 if $s$ is in the initiation set for $g$ (e.g., sufficiently close in terms of reachability) and zero otherwise. We discuss some approaches to learn this initiation function in Appendix C. But, here, we assume it is part of the discovery procedure for the subgoals and first focus on how to use it.

For the rest of this paper, we presume we are given subgoals and initiation sets. We develop algorithms to learn and use models, given those subgoals. We expect a complete agent to discover these subgoals on its own, including how to represent these subgoals to facilitate generalization and planning. To separate concerns, we focus on how the agent can leverage reasonably well-specified subgoals.

## 3.2 DEFINING SUBGOAL-CONDITIONED MODELS

For planning and acting to operate in two different spaces, we define four models: two used in planning over subgoals (subgoal-to-subgoal) and two used to project these subgoal values back into the underlying state space (state-to-subgoal). Figure 2 visualizes these two spaces.

The state-to-subgoal models are $r_\gamma : \mathcal{S} \times \bar{\mathcal{G}} \to \mathbb{R}$ and $\Gamma : \mathcal{S} \times \bar{\mathcal{G}} \to [0, 1]$, where $\bar{\mathcal{G}} = \mathcal{G} \cup \{s_{\text{terminal}}\}$ if there is a terminal state (episodic problems) and otherwise $\bar{\mathcal{G}} = \mathcal{G}$. An option policy $\pi_g : \mathcal{S} \times \mathcal{A} \to [0, 1]$ for subgoal $g$ starts from any $s$ in the initiation set, and terminates in $g$—in $\tilde{s}$ where $m(\tilde{s}, g) = 1$. The reward-model $r_\gamma(s, g)$ is the discounted rewards under option policy $\pi_g$:

$$r_\gamma(s, g) = \mathbb{E}_{\pi_g}[R_{t+1} + \gamma_g(S_{t+1})r_\gamma(S_{t+1}, g)|S_t = s]$$

where the discount is zero upon reaching subgoal $g$

$$\gamma_g(S_{t+1}) \overset{\text{def}}{=} \begin{cases} 0 & \text{if } m(S_{t+1}, g) = 1, \text{ namely if subgoal } g \text{ is achieved by being in } S_{t+1} \\ \gamma_{t+1} & \text{else} \end{cases}$$

The discount-model $\Gamma(s, g)$ reflects the discounted number of steps until reaching subgoal $g$ starting from $s$, in expectation under option policy $\pi_g$

$$\Gamma(s, g) = \mathbb{E}_{\pi_g}[m(S_{t+1}, g)\gamma_{t+1} + \gamma_g(S_{t+1})\Gamma(S_{t+1}, g)|S_t = s].$$

These state-to-subgoal will only be queried for $(s, g)$ where $d(s, g) > 0$: they are local models.

To define subgoal-to-subgoal models,[1] $\tilde{r}_\gamma : \mathcal{G} \times \bar{\mathcal{G}} \to \mathbb{R}$ and $\tilde{\Gamma} : \mathcal{G} \times \bar{\mathcal{G}} \to [0, 1]$, we use the state-to-subgoal models. For each subgoal $g \in \mathcal{G}$, we aggregate $r_\gamma(s, g')$ for all $s$ where $m(s, g) = 1$.

$$\tilde{r}_\gamma(g, g') \overset{\text{def}}{=} \frac{1}{z(g)} \sum_{s:m(s,g)=1} r_\gamma(s, g') \quad \text{and} \quad \tilde{\Gamma}(g, g') \overset{\text{def}}{=} \frac{1}{z(g)} \sum_{s:m(s,g)=1} \Gamma(s, g') \tag{1}$$

for normalizer $z(g) \overset{\text{def}}{=} \sum_{s:m(s,g)=1} m(s, g)$. This definition assumes a uniform weighting over the states $s$ where $m(s, g) = 1$. We could allow a non-uniform weighting, potentially based on visitation frequency in the environment. For this work, however, we assume that $m(s, g) = 1$ for a smaller number of states $s$ with relatively similar $r_\gamma(s, g')$, making a uniform weighting reasonable.

These models are also local models, as we can similarly extract $\tilde{d}(g, g')$ from $d(s, g')$ and only reason about $g'$ nearby or relevant to $g$. We set $\tilde{d}(g, g') = \max_{s \in \mathcal{S}:m(s,g)>0} d(s, g')$, indicating that if there is a state $s$ that is in the initiation set for $g'$ and has membership in $g$, then $g'$ is also relevant to $g$.

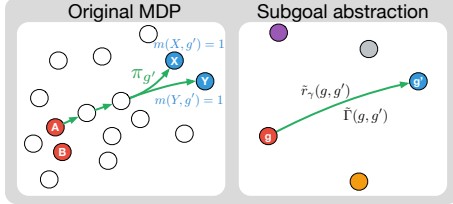

Figure 2: Original and Abstract Space.

Let us consider an example, in Figure 2. The red states are members of $g$ ($m(A, g) = 1$) and the blue members of $g'$ ($m(X, g') = 1$, $m(Y, g') = 1$). For all $s$ in the diagram, $d(s, g') > 0$ (all are in the initiation set): the policy $\pi_{g'}$ can be queried from any $s$ to get to $g'$. The green path in the left indicates the trajectory under $\pi_{g'}$ from $A$, stochastically reaching either $X$ or $Y$, with accumulated reward $r_\gamma(A, g')$ and discount $\Gamma(A, g')$ (averaged over reaching $X$ and $Y$). The subgoal-to-subgoal models, on the right, indicate $g'$ can be reached from $g$, with $\tilde{r}_\gamma(g, g')$ averaged over both $r_\gamma(A, g')$ and $r_\gamma(B, g')$ and $\tilde{\Gamma}(g, g')$ over $\Gamma(A, g')$ and $\Gamma(B, g')$, described in Equation (1).

---

[1]The first input is any $g \in \mathcal{G}$, the second is $g' \in \bar{\mathcal{G}}$, which includes $s_{\text{terminal}}$. We need to reason about reaching any subgoal or $s_{\text{terminal}}$. But $s_{\text{terminal}}$ is not a real state: we do not reason about starting from it to reach subgoals.

### 3.3 GOAL-SPACE PLANNING WITH SUBGOAL-CONDITIONED MODELS

We can now consider how to plan with these models. Planning involves learning $\tilde{v}(g)$: the value for different subgoals. This can be achieved using an update similar to value iteration, for all $g \in \mathcal{G}$

$$\tilde{v}(g) = \max_{g' \in \bar{\mathcal{G}}:\tilde{d}(g,g')>0} \tilde{r}_\gamma(g,g') + \tilde{\Gamma}(g,g')\tilde{v}(g') \quad \text{(Background Planning)} \qquad (2)$$

The value of reaching $g'$ from $g$ is the discounted rewards along the way, $\tilde{r}_\gamma(g,g')$, plus the discounted value in $g'$. If $\tilde{\Gamma}(g,g')$ is very small, it is difficult to reach $g'$ from $g$—or takes many steps—and so the value in $g'$ is discounted by more. With a relatively small number of subgoals, we can sweep through them all to quickly compute $\tilde{v}(g)$. With a larger set of subgoals, we can instead do as many updates possible, in the background on each step, by stochastically sampling $g$.

We can interpret this update as a standard value iteration update in a new MDP, where 1) the set of states is $\mathcal{G}$, 2) the actions from $g \in \mathcal{G}$ are state-dependent, corresponding to choosing which $g' \in \bar{\mathcal{G}}$ to go to in the set where $\tilde{d}(g,g') > 0$ and 3) the rewards are $\tilde{r}_\gamma$ and the discounted transition probabilities are $\tilde{\Gamma}$. Under this correspondence, it is straightforward to show that the above converges to the optimal values in this new Goal-Space MDP, shown in **Proposition 2** in Appendix B.

This goal-space planning approach does not suffer from typical issues with model-based RL. First, the model is not iterated, but we still obtain temporal abstraction because the model itself incorporates it. Second, we do not need to predict entire state vectors—or distributions over them—because we instead input the outcome $g'$ into the function approximator. This may feel like a false success as it potentially requires restricting ourselves to a smaller number of subgoals. If we want to use a larger number of subgoals, then we may need a function to generate these subgoal vectors anyway—bringing us back to the problem of generating vectors. However, this is likely easier as 1) the subgoals themselves can be much smaller and more abstract, making it more feasibly to procedurally generate them and 2) it may be more feasible maintain a large set of subgoal vectors, or generate individual subgoal vectors, than producing relevant subgoal vectors from a given subgoal.

Now let us examine how to use $\tilde{v}(g)$ to update our main policy. The simplest way to decide how to behave from a state is to cycle through the subgoals, and pick the one with the highest value.

$$v_{\text{sub}}(s) \overset{\text{def}}{=} \max_{g \in \bar{\mathcal{G}}:d(s,g)>0} r_\gamma(s,g) + \Gamma(s,g)\tilde{v}(g) \quad \text{(Projection Step)} \qquad (3)$$

and take action $a$ that corresponds to the action given by $\pi_g$ for this maximizing $g$. However, this approach has two issues. First restricting to going through subgoals might result in suboptimal policies. From a given state $s$, the set of relevant subgoals $g$ may not be on the optimal path. Second, the learned models themselves may have inaccuracies, or planning may not have been completed in the background, resulting in $\tilde{v}(g)$ that are not yet fully accurate.

We instead propose to use $v_{\text{sub}}(s)$ within the bootstrap target for the action-values for the main policy. For a given transition $(S_t, A_t, R_{t+1}, S_{t+1})$, either as the most recent experience or from a replay buffer, the proposed *subgoal-value bootstrapping* update to parameterized $q(S_t, A_t; \mathbf{w})$ uses TD error

$$\delta \overset{\text{def}}{=} R_{t+1} + \gamma_{t+1}\Big((1-\beta)\underbrace{\max_{a'} q(S_{t+1}, a'; \mathbf{w})}_{\text{Standard bootstrap target}} + \beta \underbrace{v_{\text{sub}}(S_{t+1})}_{\text{Subgoal value}}\Big) - q(S_t, A_t; \mathbf{w}) \qquad (4)$$

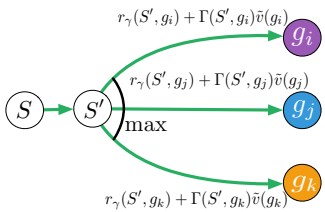

for some $\beta \in [0, 1]$. For $\beta = 0$, we get a standard Q-learning update. For $\beta = 1$, we fully bootstrap off the value provided by $v_{\text{sub}}(S_{t+1})$. This may result in suboptimal values $q(S_t, A_t; \mathbf{w})$, but should learn faster because a reasonable estimate of value has been propagated back quickly using goal-space planning. On the other hand, $\beta = 0$ is not biased by a potentially suboptimal $\tilde{v}(g)$, but does not take advantage of this fast propagation. An interim $\beta$ can allow for fast propagation, but also help overcome suboptimality in the values.

Figure 3: Computing $v_{\text{sub}}(S')$ to update the policy at $S$.

We can show that the above update improves the convergence rate. This result is intuitive: subgoal-value bootstrapping changes the discount rate to $\gamma_{t+1}(1 - \beta)$. In the extreme case of $\beta = 1$, we are moving our estimate towards $R_{t+1} + \gamma_{t+1}v_{\text{sub}}(S_{t+1})$ for $v_{\text{sub}}$ not based on $q$ without any bootstrapping: it is effectively a regression problem. We prove this intuitive result in **Proposition 3** in Appendix B. One other benefit of this approach is that the initiation sets need not cover the whole space: we can have a state $d(s, g) = 0$ for all $g$. If this occurs, we simply do not use $v_{\text{sub}}$ and bootstrap as usual.

### 3.4 Putting it All Together: The Full Goal-Space Planning Algorithm

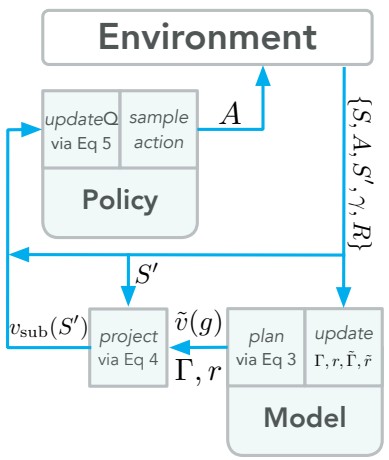

Figure 4: Goal-Space Planning.

The remaining piece is to learn the models and put it all together. Learning the models is straightforward, as we can leverage the large literature on general value functions (Sutton et al., 2011) and UVFAs (Schaul et al., 2015). There are nuances involved in 1) restricting updating to relevant states according to $d(s, g)$, 2) learning option policies that reach subgoals, but also maximize rewards along the way and 3) considering ways to jointly learn $d$ and $\Gamma$. For space we include these details in Appendix C.

The algorithm is visualized in Figure 4 (pseudocode in appx. C.3). The steps of agent-environment interaction include:
1) take action $A_t$ in state $S_t$, to get $S_{t+1}, R_{t+1}$ and $\gamma_{t+1}$;
2) query the model for $r_\gamma(S_{t+1}, g), \Gamma(S_{t+1}, g), \tilde{v}(g)$ for all $g$
   where $d(S_{t+1}, g) > 0$;
3) compute projection $v_{\text{sub}}(S_{t+1})$ using Eq. (3) and step 2;
4) update the main policy with the transition and $v_{\text{sub}}(S_{t+1})$,
   using Eq. (4).

All background computation is used for model learning using a replay buffer and for planning to obtain $\tilde{v}$, so that they can be queried at any time on step 2.

## 4 Experiments with Goal-Space Planning

We investigate the utility of GSP, for 1) improving sample efficiency and 2) re-learning under non-stationarity. We compare to Double DQN (DDQN) (Van Hasselt et al., 2016), which uses replay and target networks. We layer GSP on top of this agent: the action-value update is modified to incorporate subgoal-value bootstrapping. By selecting $\beta = 0$, we perfectly recover DDQN, allowing us to test different $\beta$ values to investigate the impact of incorporating subgoal values computed using background planning.

### 4.1 Experiment Specification

We test the agents in the PinBall environment (Konidaris & Barto, 2009), which allows for a variety of easy and harder instances to test different aspects. The agent has to navigate a small ball to a destination in a maze-like environment with fully elastic and irregularly shaped obstacles. The state is described by 4 features: $x \in [0, 1], y \in [0, 1], \dot{x} \in [-1, 1], \dot{y} \in [-1, 1]$. The agent has 5 discrete actions: increase/decrease $\dot{x}$, increase/decrease $\dot{y}$, and nothing. The agent receives a reward of -5 per step and a reward of 10,000 upon termination at the goal location. PinBall has a continuous state space with complex and sharp dynamics that make learning and control difficult. We used a harder version of PinBall in our first experiment, shown in Figure 5, and simpler one for the non-stationary experiment, shown in Figure 9, to allow DDQN a better chance to adapt under non-stationarity.

The hyperparameters are chosen based on sweeping for DDQN performance. We then fixed these hyperparameters, and used them for GSP. This approach helps ensure they have similar settings, with the primary difference due to incorporating subgoal-value bootstrapping. We used neural networks with ReLU activations and $\epsilon = 0.1$; details about hyperparameters are in Appendix F.

The set of subgoals for GSP are chosen to cover the environment in terms of $(x, y)$ locations. For each subgoal $g$ with location $(x_g, y_g)$, we set $m(s, g) = 1$ for $s = (x, y, \dot{x}, \dot{y})$ if the Euclidean distance between $(x, y)$ and $(x_g, y_g)$ is below 0.035. Using a region, rather than requiring $(x, y) = (x_g, y_g)$, is necessary for a continuous state space. The agent's velocity is not taken into account for subgoal termination. The width of the region for the initiation function is 0.4. More details about the layout of the environment, positions of these subgoals and initiation functions are shown in Figure 5.

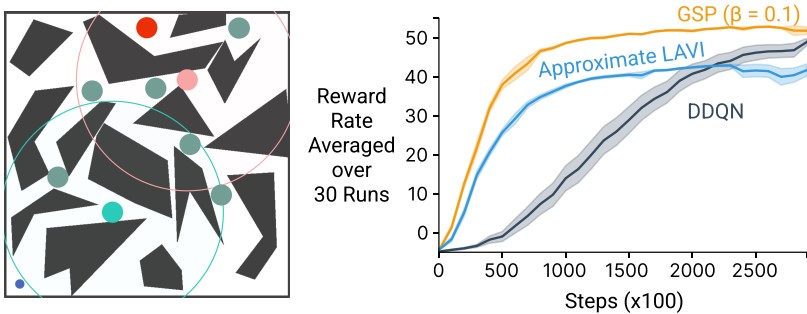

Figure 5: **(left)** The harder PinBall environment used in our first experiment. The dark gray shapes are obstacles the ball bounces off of, the small blue circle the starting position of the ball (with no velocity), and the red dot the goal (termination). Solid circles indicate the location and radius of the subgoals ($m$), with wider initiation set visualized for two subgoals (pink and teal). **(right)** Performance in this environment for GSP with $\beta = 0.1$, DDQN, and approximate LAVI, with the standard error shown. Even just increasing to $\beta = 0.1$ allows GSP to leverage the longer-horizon estimates given by the subgoal values, making it learn much faster than DDQN. Approximate LAVI is able to learn quickly, but levels off at a suboptiomal performance, as expected.

## 4.2 EXPERIMENT 1: INVESTIGATING GSP WITH PRE-TRAINED MODELS

We first investigate the utility of the models after they have been learned in a pre-training phase. The models use the same updates as they would when being learned online, and are not perfectly accurate. Pre-training the model allows us to ask: if the GSP agent had previously learned a model in the environment—or had offline data to train its model—can it leverage it to learn faster now? One of the primary goals of model-based RL is precisely this re-use, and so it is natural to start in a setting mimicking this use-case. We assume the GSP agent can do many steps of background planning, so that $\tilde{v}$ is effectively computed in early learning; this is reasonable as we only need to do value iteration for 9 subgoals, which is fast. We compare GSP with $\beta = 0.1$ against two baselines: DDQN and approximate LAVI. DDQN is the model-free baseline which GSP builds on top of, and can also be viewed as a version of Dyna when the replay buffer is viewed as a non-parametric model in the PinBall environment (van Hasselt et al., 2019; Pan et al., 2018), with planning updates sampled based on the agent's prior state-action visitation distribution. Approximate LAVI is a modified version of LAVI (Mann et al., 2015) that uses learned subgoal models, and is a version of GSP that fully relies on subgoal values when performing its updates with $\beta = 0$. We selected $\beta = 0.1$ as it provided the best tradeoff of the beta values but we find that $\beta$ as small as $1e^{-3}$ was able to outperform DDQN. The performance of GSP with different $\beta$ can be found in Appendix I.

We see in Figure 5 that GSP learns much faster than DDQN, and reaches the same level of performance. This is the result we should expect—GSP gets to leverage a pre-trained model, after all—but it is an important sanity check that using models in this new way is effective. Of particular note is that even just increasing $\beta$ from 0 (which is DDQN) to $\beta = 0.1$ provides the learning speed boost without resulting in suboptimal performance. Likely, in early learning, the suboptimal subgoal values provide a coarse direction to follow, to more quickly update the action-values, which is then refined with more learning. When the approximate subgoal models are fully relied upon on as with approximate LAVI, we similarly get fast initial learning, but it plateaus at a more suboptimal point.

To further investigate the hypothesis that GSP more quickly changes its value function early in learning, we visualize the value functions for both GSP and DDQN over time in Figure 6. After 2000 steps, they are not yet that different, because there are only four replay updates on each step and it takes time to visit the state-space and update values by bootstrapping off of subgoal values. By step 6000, though, GSP already has some of the structure of the problem, whereas DDQN has simply pushed down many of its values (darker blue).

To see whether GSP is feasible to apply to other problems, we also evaluated GSP in Lunar Lander (Brockman et al., 2016), an environment where subgoal specification is not as obvious and environment dynamics cause the agent to frequently crash. We include those results in Appendix H, but note that similar conclusions about comparisons between GSP and DDQN hold. We also compared GSP to various Dyna-style planning algorithms, some of which also incorporates temporal abstraction, in Appendix G and find that GSP is able to outperform these alternatives.

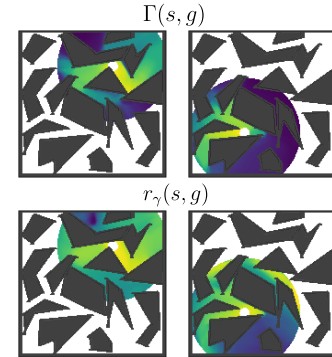

Figure 6: **(left)** Visualizing the action-values for DDQN and GSP ($\beta = 0.1$) at various points in training. **(right)** $v_{\text{sub}}$ obtained from using the learned subgoal-values in the projection step.

## 4.3 Accuracy of the Learned Models

One potential benefit of GSP is that the models themselves may be easier to learn, because we can leverage standard value function learning algorithms. We visualize the models learned for the previous experiment, as well as the resulting $v_{\text{sub}}$, with details about model learning in Appendix E.

In Figure 7 we see how learned state-to-subgoal models accurately learn the structure. Each plot shows the learned state-to-subgoal for one subgoal, visualized only for the initiation set $d(s, g) > 0$. We can see larger discount and reward values predicted based on reachability. However, the models are not perfect. We measured model error and find it is reasonable but not very near zero (see Appendix E). This result is actually encouraging: inaccuracies in the model do not prevent useful planning.

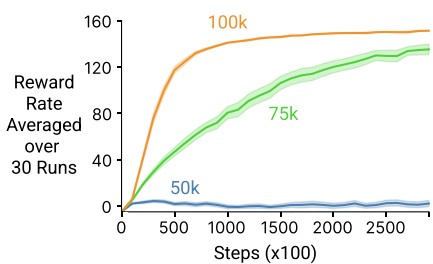

Figure 7: Learned state-to-subgoal models. White indicates $d(s, g) = 0$.

It is informative to visualize $v_{\text{sub}}$. We can see in Figure 6 that the general structure is correct, matching the optimal path, but that it indeed looks suboptimal compared to the final values computed in Figure 6 by DDQN. This inaccuracy is likely due both to some inaccuracy in the models, as well as the fact that subgoal placement is not optimal. This explains why GSP has lower values particularly in states near the bottom, likely skewed downwards by $v_{\text{sub}}$.

Finally, we test the impact on learning using less accurate models. After all, the agent will want to start using its model as soon as possible, rather than waiting for it to become more accurate. We ran GSP using models learned online, using only 50k, 75k and 100k time steps to learn the models. We then froze the models and allowed GSP to learn with them. We can see in Figure 8 that learning with too inaccurate of a model—with 50k—fails, but already with 75k performance improves considerably and with 100k we are already nearly at the same level of optimal performance as the pre-trained models. This result highlights it should be feasible to learn and use these models in GSP, all online.

Figure 8: The impact on planning performance using frozen models with differing accuracy (shading shows standard error).

## 4.4 Experiment 2: Adapting in Nonstationary PinBall

Now we consider another typical use-case for model-based RL: quickly adapting to changes in the environment. We let the agent learn in PinBall for 50k steps, and then switch the goal to a new location for another 50k steps. Goal information is never given to the agent, so it has to visit the old goal, realize it is no longer rewarding, and re-explore to find the new goal. This non-stationary setting is harder for DDQN, so we use a simpler configuration for PinBall, shown in Figure 9.

We can leverage the idea of exploration bonuses, introduced in Dyna-Q+ (Sutton & Barto, 2018). Exploration bonuses are proportional to the last time that state-action was visited. This encourages the agent to revisit parts of the state-space that it has not seen recently, in case that part of the world has changed. For us, this corresponds to including reward bonus $r_{\text{bonus}}$ in the planning and projection steps: $\tilde{v}(g) = \max_{g' \in \bar{\mathcal{G}}: \tilde{d}(g,g') > 0} \tilde{r}_\gamma(g, g') + \tilde{\Gamma}(g, g') \left( \tilde{v}(g') + r_{\text{bonus}}(g') \right)$ and $v_{\text{sub}}(s) =$

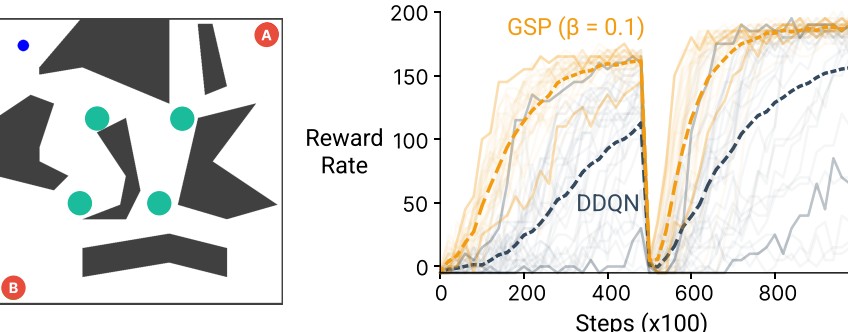

Figure 9: **(left)** The Non-stationary PinBall environment. For the first half of the experiment, the agent terminates at goal A while for the second half, the agent terminates at goal B. **(right)** The performance of GSP ($\beta = 0.1$) and DDQN in the environment. The mean of all 30 runs is shown as the dashed line. The 25th and 75th percentile run for each algorithm are also highlighted. We see that GSP with exploration bonus was able to adapt more quickly when the terminal goal switches compared to the baseline DDQN algorithm where goal values are not used.

$\max_{g \in \bar{\mathcal{G}}:d(s,g)>0} r_\gamma(s,g) + \Gamma(s,g) \left(\tilde{v}(g) + r_{\text{bonus}}(g)\right)$. Because we have a small, finite set of subgoals, it is straightforward to leverage this idea that was designed for the tabular setting. We use $r_{\text{bonus}}(g) = 1000$ if the count for $g$ is zero, and 0 otherwise. When the world changes, the agent recognizes that it has changed, and resets all counts. Similarly, both agents (GSP and DDQN) clear their replay buffers.

The GSP agent can recognize the world has changed, but not how it has changed. It has to update its models with experience. The state-to-subgoal models and subgoal-to-subgoal models local to the previous terminal state location and the new one need to change, but the rest of the models are actually already accurate. The agent can leverage this existing accuracy.

In Figure 9, we can see both GSP and DDQN drop in performance when the environment changes, with GSP recovering much more quickly. It is always possible that an inaccurate model might actually make re-learning slower, reinforcing incorrect values from the model. Here, though, updating these local models is fast, allowing the subgoal values to also be updated quickly. Though not shown in the plot, GSP without exploration bonuses performs poorly. Its model causes it to avoid visiting the new goal region, so preventing the model from updating, because the value in that bottom corner is low.

## 5 CONCLUSION

In this paper we introduced a new planning framework, called Goal-Space Planning (GSP). GSP provides a new approach to use background planning to improve action-value estimates, with minimalist, local models and computationally efficient planning. We show in the PinBall environment that these subgoal-conditioned models can be accurately learned using standard value estimation algorithms and that GSP is robust to less accurate models (Section 4.3). We also find that GSP can significantly improve the speed of learning over DDQN in both the PinBall environment and outperforms several Dyna variants, including Dyna with options (Appendix G), and that GSP relearns more quickly under non-stationarity than DDQN (Section 4.4). Additionally, we compared GSP to DDQN in another environment, called Lunar Lander (Appendix H), both to highlight that the conclusions extend and to demonstrate that it is straightforward to apply GSP to other problems.

This work introduces a new formalism, and many new technical questions along with it. We have only tested GSP with pre-trained models and assumed a given set of subgoals. Our initial experiments learning the models online, from scratch, indicate that GSP can get similar learning speed boosts. Using a recency buffer, however, accumulates transitions only along the optimal trajectory, sometimes causing the models to become inaccurate part-way through learning. An important next step is to incorporate smarter model learning strategies. The other critical open question is in subgoal discovery. We somewhat randomly selected subgoals across the PinBall environment, with a successful outcome; such an approach is unlikely to work in many environments. In general, option discovery and subgoal discovery remain open questions. One utility of this work is that it could help narrow the scope of the discovery question, to that of finding abstract subgoals that help the agent plan more efficiently.

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

## A  STARTING SIMPLER: GOAL-SPACE PLANNING FOR POLICY EVALUATION

To highlight the key idea for efficient planning, we provide an example of GSP in a simpler setting: policy evaluation for learning $v^\pi$ for a fixed deterministic policy $\pi$ in a deterministic environment, assuming access to the true models. The key idea is to propagate values quickly across the space by updating between a subset of states that we call *subgoals*, $g \in \mathcal{G} \subset \mathcal{S}$, as visualized in Figure 10. (Later we extend $\mathcal{G} \not\subset \mathcal{S}$ to abstract subgoal vectors that need not correspond to any state.) To do so, we need temporally extended models between pairs $g, g'$ that may be further than one-transition apart. For policy evaluation, these models are the accumulated rewards $r_{\pi,\gamma} : \mathcal{S} \times \mathcal{S} \to \mathbb{R}$ and discounted probabilities $P_{\pi,\gamma} : \mathcal{S} \times \mathcal{S} \to [0, 1]$ under $\pi$:

$$r_{\pi,\gamma}(g, g') \stackrel{\text{def}}{=} \mathbb{E}_\pi[R_{t+1} + \gamma_{g',t+1} r_{\pi,\gamma}(S_{t+1}, g')|S_t = g]$$

$$P_{\pi,\gamma}(g, g') \stackrel{\text{def}}{=} \mathbb{E}_\pi[1(S_{t+1} = g')\gamma_{t+1} + \gamma_{g',t+1} P_{\pi,\gamma}(S_{t+1}, g')|S_t = g]$$

where $\gamma_{g',t+1} = 0$ if $S_{t+1} = g'$ and otherwise equals $\gamma_{t+1}$, the environment discount. If we cannot reach $g'$ from $g$ under $\pi$, then $P_{\pi,\gamma}(g, g')$ will simply accumulate many zeros and be zero. We can treat $\mathcal{G}$ as our new state space and plan in this space, to get value estimates $v$ for all $g \in \mathcal{G}$

$$v(g) = r_{\pi,\gamma}(g, g') + P_{\pi,\gamma}(g, g')v(g') \quad \text{where } g' = \text{argmax}_{g' \in \bar{\mathcal{G}}} P_{\pi,\gamma}(g, g')$$

where $\bar{\mathcal{G}} = \mathcal{G} \cup \{s_{\text{terminal}}\}$ if there is a terminal state (episodic problems) and otherwise $\bar{\mathcal{G}} = \mathcal{G}$. It is straightforward to show this converges, because $P_{\pi,\gamma}$ is a substochastic matrix (see Appendix A.1).

Once we have these values, we can propagate these to other states, locally, again using the closest $g$ to $s$. We can do so by noticing that the above definitions can be easily extended to $r_{\pi,\gamma}(s, g')$ and $P_{\pi,\gamma}(s, g')$, since for a pair $(s, g)$ they are about starting in the state $s$ and reaching $g$ under $\pi$.

$$v(s) = r_\gamma(s, g) + P_{\pi,\gamma}(s, g)v(g) \quad \text{where } g = \text{argmax}_{g \in \bar{\mathcal{G}}} P_{\pi,\gamma}(s, g). \tag{5}$$

Because the rhs of this equation is fixed, we only cycle through these states once to get their values.

All of this might seem like a lot of work for policy evaluation; indeed, it will be more useful to have this formalism for control. But, even here goal-space planning can be beneficial. Let assume a chain $s_1, s_2, \ldots, s_n$, where $n = 1000$ and $\mathcal{G} = \{s_{100}, s_{200}, \ldots, s_{1000}\}$. Planning over $g \in \mathcal{G}$ only requires sweeping over 10 states, rather than 1000. Further, we have taken a 1000 horizon problem and converted it into a 10 step one.[2] As a result, changes in the environment also propagate faster. If the reward at $s'$ changes, locally the reward model around $s'$ can be updated quickly, to change $r_{\pi,\gamma}(g, g')$ for pairs $g, g'$ where $s'$ is along the way from $g$ to $g'$. This local change quickly updates the values back to earlier $\tilde{g} \in \mathcal{G}$.

### A.1  PROOFS FOR THE DETERMINISTIC POLICY EVALUATION SETTING

We provide proofs here for the deterministic policy evaluation setting. We assume throughout that the environment discount $\gamma_{t+1}$ is a constant $\gamma_c \in [0, 1)$ for every step in an episode, until termination when it is zero. The below results can be extended to the case where $\gamma_c = 1$, using the standard strategy for the stochastic shortest path problem setting.

First, we want to show that given $r_{\pi,\gamma}$ and $P_{\pi,\gamma}$, we can guarantee that the update for the values for $\mathcal{G}$ will converge. Recall that $\bar{\mathcal{G}} = \mathcal{G} \cup \{s_{\text{terminal}}\}$ is the augmented goal space that includes the terminal state. This terminal state is not a subgoal—since it is not a real state—but is key for appropriate planning.

**Lemma 1.** *Assume that we have a deterministic MDP, deterministic policy $\pi$, $\gamma_c < 1$, a discrete set of subgoals $\mathcal{G} \subset \mathcal{S}$, and that we iteratively update $v_t \in \mathbb{R}^{|\mathcal{G}|}$ with the dynamic programming update*

$$v_t(g) = r_{\pi,\gamma}(g, g') + P_{\pi,\gamma}(g, g')v_{t-1}(g') \quad \text{where } g' = \text{argmax}_{g' \in \bar{\mathcal{G}}} P_{\pi,\gamma}(g, g') \tag{6}$$

*for all $g \in \mathcal{G}$, starting from an arbitrary (finite) initialization $v_0 \in \mathbb{R}^{|\mathcal{G}|}$, with $v_t(s_{terminal})$ fixed at zero. Then then $v_t$ converges to a fixed point.*

---

[2]In this simplified example, we can plan efficiently by updating the value at the end in $s_n$, and then updating states backwards from the end. But, without knowing this structure, it is not a general purpose strategy. For general MDPs, we would need smart ways to do search control: the approach to pick states from one-step updates. In fact, we can leverage search control strategies to improve the goal-space planning step. Then we get the benefit of these approaches, as well as the benefit of planning over a much smaller state space.

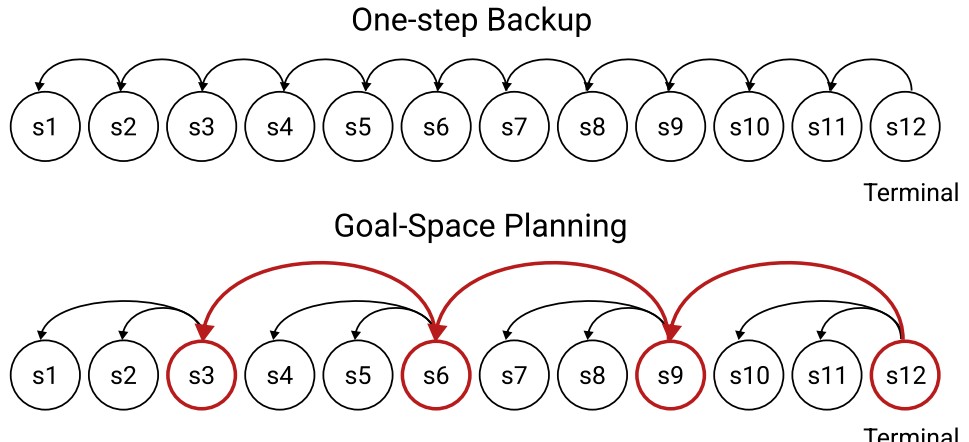

Figure 10: Comparing one-step backup with Goal-Space Planning when subgoals are concrete states. GSP first focuses planning over a smaller set of subgoals (in red), then updates the values of individual states.

*Proof.* To analyze this as a matrix update, we need to extend $P_{\pi,\gamma}(g, g')$ to include an additional row transitioning from $s_{\text{terminal}}$. This row is all zeros, because the value in the terminal state is always fixed at zero. Note that there are ways to avoid introducing terminal states, using transition-based discounting (White, 2017), but for this work it is actually simpler to explicitly reason about them and reaching them from subgoals.

To show this we simply need to ensure that $P_{\pi,\gamma}$ is a substochastic matrix. Recall that

$$P_{\pi,\gamma}(g, g') \overset{\text{def}}{=} \mathbb{E}_\pi[1(S_{t+1} = g')\gamma_{t+1} + \gamma_{g',t+1}P_{\pi,\gamma}(S_{t+1}, g')|S_t = g]$$

where $\gamma_{g',t+1} = 0$ if $S_{t+1} = g'$ and otherwise equals $\gamma_{t+1}$, the environment discount. If it is substochastic, then $\|P_{\pi,\gamma}\|_2 < 1$. Consequently, the Bellman operator

$$(Bv)(g) = r_{\pi,\gamma}(g, g') + P_{\pi,\gamma}(g, g')\tilde{v}(g') \quad \text{where } g' = \underset{g' \in \bar{\mathcal{G}}}{\arg\max}\, P_{\pi,\gamma}(g, g')$$

is a contraction, because $\|Bv_1 - Bv_2\|_2 = \|P_{\pi,\gamma}v_1 - P_{\pi,\gamma}v_2\|_2 \le \|P_{\pi,\gamma}\|_2\|v_1 - v_2\|_2 < \|v_1 - v_2\|_2$.

Because $\gamma_c < 1$, then either $g$ immediately terminates in $g'$, giving $1(S_{t+1} = g')\gamma_{t+1} + \gamma_{g',t+1}P_{\pi,\gamma}(S_{t+1}, g') = \gamma_{t+1} + 0 \le \gamma_c$. Or, it does not immediately terminate, and $1(S_{t+1} = g')\gamma_{t+1} + \gamma_{g',t+1}P_{\pi,\gamma}(S_{t+1}, g') = 0 + \gamma_c P_{\pi,\gamma}(S_{t+1}, g') \le \gamma_c$ because $P_{\pi,\gamma}(S_{t+1}, g') \le 1$. Therefore, if $\gamma_c < 1$, then $\|P_{\pi,\gamma}\|_2 \le \gamma_c$.

$\square$

**Proposition 1.** *For a deterministic MDP, deterministic policy $\pi$, and a discrete set of subgoals $\mathcal{G} \subset \mathcal{S}$ that are all reached by $\pi$ in the MDP, given the $\tilde{v}(g)$ obtained from Equation 6, if we set*

$$v(s) = r_\gamma(s, g) + P_{\pi,\gamma}(s, g)\tilde{v}(g) \quad \text{where } g = \underset{g \in \bar{\mathcal{G}}}{\arg\max}\, P_{\pi,\gamma}(s, g) \tag{7}$$

*for all states $s \in \mathcal{S}$ then we get that $v = v_\pi$.*

*Proof.* For a deterministic environment and deterministic policy this result is straightforward. The term $P_{\pi,\gamma}(s, g) > 0$ only if $g$ is on the trajectory from $s$ when the policy $\pi$ is executed. The term $r_\gamma(s, g)$ consists of deterministic (discounted) rewards and $\tilde{v}(g)$ is the true value from $g$, as shown in Lemma 6 (namely $\tilde{v}(g) = v_\pi(g)$). The subgoal $g$ is the closest state on the trajectory from $s$, and $P_{\pi,\gamma}(s, g)$ is $\gamma_c^t$ where $t$ is the number of steps from $s$ to $g$.

$\square$

## B  PROOFS FOR THE GENERAL CONTROL SETTING

In this section we assume that $\gamma_c < 1$, to avoid some of the additional issues for handling proper policies. The same strategies apply to the stochastic shortest path setting with $\gamma_c = 1$, with additional assumptions.

**Proposition 2.** *[Convergence of Value Iteration in Goal-Space] Assuming that $\tilde{\Gamma}$ is a substochastic matrix, with $v_0 \in \mathbb{R}^{|\mathcal{G}|}$ initialized to an arbitrary value and fixing $v_t(s_{terminal}) = 0$ for all t, then iteratively sweeping through all $g \in \mathcal{G}$ with update*

$$v_t(g) = \max_{g' \in \bar{\mathcal{G}}: \bar{d}(g,g')>0} \tilde{r}_\gamma(g,g') + \tilde{\Gamma}(g,g')v_{t-1}(g') \tag{8}$$

*convergences to a fixed-point.*

*Proof.* We can use the same approach typically used for value iteration. For any $v_0 \in \mathbb{R}^{|\mathcal{G}|}$, we can define the operator

$$(B^g v)(g) \stackrel{\text{def}}{=} \max_{g' \in \bar{\mathcal{G}}: \bar{d}(g,g')>0} \tilde{r}_\gamma(g,g') + \tilde{\Gamma}(g,g')\tilde{v}(g')$$

First we can show that $B^g$ is a $\gamma_c$-contraction. Assume we are given any two vectors $v_1, v_2$. Notice that $\tilde{\Gamma}(g,g') \leq \gamma_c$, because for our problem setting the discount is either equal to $\gamma_c$ or equal to zero at termination. Then we have that for any $g \in \bar{\mathcal{G}}$

$$|(B^g v_1)(g) - (B^g v_2)(g)|$$
$$= \left| \max_{g' \in \bar{\mathcal{G}}: \bar{d}(g,g')>0} \tilde{r}_\gamma(g,g') + \tilde{\Gamma}(g,g')v_1(g') - \max_{g' \in \bar{\mathcal{G}}: \bar{d}(g,g')>0} \tilde{r}_\gamma(g,g') + \tilde{\Gamma}(g,g')v_2(g') \right|$$
$$\leq \max_{g' \in \bar{\mathcal{G}}: \bar{d}(g,g')>0} |\tilde{r}_\gamma(g,g') + \tilde{\Gamma}(g,g')v_1(g') - (\tilde{r}_\gamma(g,g') + \tilde{\Gamma}(g,g')v_2(g'))|$$
$$= \max_{g' \in \bar{\mathcal{G}}: \bar{d}(g,g')>0} |\tilde{\Gamma}(g,g')(v_1(g') - v_2(g'))|$$
$$\leq \max_{g' \in \bar{\mathcal{G}}: \bar{d}(g,g')>0} \gamma_c |v_1(g') - v_2(g')|$$
$$\leq \gamma_c \|v_1 - v_2\|_\infty$$

Since this is true for any $g$, it is true for the max over $g$, giving

$$\|B^g v_1 - B^g v_2\|_\infty \leq \gamma_c \|v_1 - v_2\|_\infty.$$

Because the operator $B^g$ is a contraction, since $\gamma_c < 1$, we know by the Banach Fixed-Point Theorem that the fixed-point exists and is unique. □

Now we analyze the update to the main policy, that incorporates the subgoal value estimates into the bootstrap target. We assume we have a finite number of state-action pairs $n$, with parameterized action-values $q(\cdot; \mathbf{w}) \in \mathbb{R}^n$ represented as a vector with one entry per state-action pair. Value iteration to find $q^*$ corresponds to updating with the Bellman optimality operator

$$(Bq)(s,a) \stackrel{\text{def}}{=} r(s,a) + \sum_{s'} P(s'|s,a)\gamma(s') \max_{a' \in \mathcal{A}} q(s',a') \tag{9}$$

On each step, for the current $q_t \stackrel{\text{def}}{=} q(\cdot; \mathbf{w}_t)$, if we assume the parameterized function class can represent $Bq_t$, then we can reason about the iterations of $\mathbf{w}_1, \mathbf{w}_2, \dots$ obtain when minimizing distance between $q(\cdot; \mathbf{w}_{t+1})$ and $Bq_t$, with

$$q(s,a; \mathbf{w}_{t+1}) = (Bq(\cdot; \mathbf{w}_t))(s,a)$$

Under function approximation, we do not simple update a table of values, but we can get this equality by minimizing until we have zero Bellman error. Note that $q^* = Bq^*$, by definition.

In this *realizability* regime, we can reason about the iterates produced by value iteration. The convergence rate is dictated by $\gamma_c$, as is well known, because

$$\|Bq_1 - Bq_2\|_\infty \leq \gamma_c \|q_1 - q_2\|_\infty$$

Specifically, if we assume $|r(s, a)| \leq r_{\max}$, then we can use the fact that 1) the maximal return is no greater than $G_{\max} \stackrel{\text{def}}{=} \frac{r_{\max}}{1-\gamma_c}$, and 2) for any initialization $q_0$ no larger in magnitude than this maximal return we have that $\|q_0 - q^*\|_\infty \leq 2G_{\max}$. Therefore, we get that

$$\|Bq_0 - q^*\|_\infty = \|Bq_0 - Bq^*\|_\infty \leq \gamma_c \|q_0 - q^*\|_\infty$$

and so after $t$ iterations we have

$$\|q_t - q^*\|_\infty = \|Bq_{t-1} - Bq^*\|_\infty \leq \gamma_c \|q_{t-1} - q^*\|_\infty \leq \gamma_c^2 \|q_{t-2} - q^*\|_\infty \cdots \leq \gamma_c^t \|q_0 - q^*\|_\infty = \gamma_c^t G_{\max}$$

We can use the exact same strategy to show convergence of value iteration, under our subgoal-value bootstrapping update. Let $r_{\text{sub}}(s, a) \stackrel{\text{def}}{=} \sum_{s'} P(s'|s, a)v_{\text{sub}}(s')$, assuming $v_{\text{sub}} : \mathcal{S} \to [-G_{\max}, G_{\max}]$ is a given, fixed function. Then the modified Bellman optimality operator is

$$(B^\beta q)(s, a) \stackrel{\text{def}}{=} r(s, a) + \beta r_{\text{sub}}(s, a) + (1-\beta) \sum_{s'} P(s'|s, a)\gamma(s') \max_{a' \in \mathcal{A}} q(s', a') \qquad (10)$$

**Proposition 3** (Convergence rate of tabular value iteration under subgoal bootstrapping). *The fixed point $q_\beta^* = B^\beta q_\beta^*$ exists and is unique. Further, for $q_0$, and the corresponding $\mathbf{w}_0$, initialized such that $|q_0(s, a; \mathbf{w}_0)| \leq G_{max}$, the value iteration update with subgoal bootstrapping $q_t = B^\beta q_{t-1}$ for $t = 1, 2, \ldots$ satisfies*

$$\|q_t - q_\beta^*\|_\infty \leq (1-\beta)^t \gamma_c^t \frac{r_{max} + \beta G_{max}}{1 - (1-\beta)\gamma_c}$$

*Proof.* First we can show that $B^\beta$ is a $\gamma_c(1-\beta)$-contraction. Assume we are given any two vectors $q_1, q_2$. Notice that $\gamma(s) \leq \gamma_c$, because for our problem setting it is either equal to $\gamma_c$ or equal to zero at termination. Then we have that for any $(s, a)$

$$|(B^\beta q_1(s, a) - (B^\beta q_2)(s, a)| = |(1-\beta) \sum_{s'} P(s'|s, a)\gamma(s')[\max_{a' \in \mathcal{A}} q_1(s', a') - \max_{a' \in \mathcal{A}} q_2(s', a')]|$$

$$\leq (1-\beta)\gamma_c \sum_{s'} P(s'|s, a)|[\max_{a' \in \mathcal{A}} q_1(s', a') - \max_{a' \in \mathcal{A}} q_2(s', a')]|$$

$$\leq (1-\beta)\gamma_c \sum_{s'} P(s'|s, a) \max_{a' \in \mathcal{A}} |q_1(s', a') - q_2(s', a')|$$

$$\leq (1-\beta)\gamma_c \sum_{s'} P(s'|s, a) \max_{s' \in \mathcal{S}, a' \in \mathcal{A}} |q_1(s', a') - q_2(s', a')|$$

$$\leq (1-\beta)\gamma_c \sum_{s'} P(s'|s, a)\|q_1 - q_2\|_\infty$$

$$= (1-\beta)\gamma_c \|q_1 - q_2\|_\infty$$

Since this is true for any $(s, a)$, it is true for the max, giving

$$\|B^\beta q_1 - B^\beta q_2\|_\infty \leq (1-\beta)\gamma_c \|q_1 - q_2\|_\infty.$$

Because the operator is a contraction, since $(1-\beta)\gamma_c < 1$, we know by the Banach Fixed-Point Theorem that the fixed-point exists and is unique.

Now we can also use contraction property for the convergence rate. Notice first that we can consider $\tilde{r}(s, a) \stackrel{\text{def}}{=} r(s, a) + \beta r_{\text{sub}}(s, a)$ as the new reward, with maximum value $r_{\max} + \beta G_{\max}$. Further, the new discount is $(1-\beta)\gamma_c$. Consequently, the maximal return is $\frac{r_{\max} + \beta G_{\max}}{1 - (1-\beta)\gamma_c}$.

$$\|q_t - q_\beta^*\|_\infty = \|B^\beta q_{t-1} - B^\beta q_\beta^*\|_\infty \leq (1-\beta)\gamma_c \|q_{t-1} - q^*\|_\infty \cdots \leq (1-\beta)^t \gamma_c^t \|q_0 - q^*\|_\infty$$

$$\leq (1-\beta)^t \gamma_c^t \frac{r_{\max} + \beta G_{\max}}{1 - (1-\beta)\gamma_c} \qquad \square$$

This rate is dominated by $((1-\beta)\gamma_c)^t$, and for $\beta$ near 1 gives a much faster convergence rate than $\beta = 0$. We can determine after how many iteration this term overcomes the increase in the upper bound on the return. In other words, we want to know how big $t$ needs to be to get

$$(1-\beta)^t \gamma_c^t \frac{r_{\max} + \beta G_{\max}}{1 - (1-\beta)\gamma_c} \leq \gamma_c^t G_{\max}.$$

Rearranging terms, we get that this is true for

$$t > \log\left(\frac{r_{\max} + \beta G_{\max}}{G_{\max}(1 - (1 - \beta)\gamma_c)}\right) / \log\left(\frac{1}{1 - \beta}\right).$$

For example if $r_{\max} = 1$, $\gamma_c = 0.99$ and $\beta = 0.5$, then we have that $t > 1.56$. If we have that $r_{\max} = 10$, $\gamma_c = 0.99$ and $\beta = 0.5$, then we get that $t \geq 5$. If we have that $r_{\max} = 1$, $\gamma_c = 0.99$ and $\beta = 0.1$, then we get that $t \geq 22$.

## C  LEARNING THE SUBGOAL MODELS AND CORRESPONDING OPTION POLICIES

Now we need a way to learn the models, $r_\gamma(s, g)$ and $\Gamma(s, g)$. These can both be represented as General Value Functions (GVFs) (Sutton et al., 2011), and we leverage this form to use standard algorithms in reinforcement learning to learn them. We start by assuming that we have $\pi_g$ and discuss learning it after understanding learning these models.

### C.1  MODEL LEARNING

The data is generated off-policy—according to some behavior $b$ rather than from $\pi_g$. We can either use importance sampling or we can learn the action-value variants of these models to avoid importance sampling. We describe both options here, but in our experiments using the action-value variant since it avoids importance sampling and the need to have the distribution over actions under behavior $b$.

**Model Update using Importance Sampling**  We can update $r_\gamma(\cdot, g)$ with an importance-sampled temporal difference (TD) learning update $\rho_t \delta_t \nabla r_\gamma(S_t, g)$ where $\rho_t = \frac{\pi_g(a|S_t)}{b(a|S_t)}$ and

$$\delta_t^r = R_{t+1} + \gamma_{g,t+1} r_\gamma(S_{t+1}, g) - r_\gamma(S_t, g)$$

The discount model $\Gamma(s, g)$ can be learned similarly, because it is also a GVF with cumulant $m(S_{t+1}, g)\gamma_{t+1}$ and discount $\gamma_{g,t+1}$. The TD update is $\rho_t \delta_t^\Gamma$ where

$$\delta_t^\Gamma = m(S_{t+1}, g)\gamma_{t+1} + \gamma_{g,t+1}\Gamma(S_{t+1}, g) - \Gamma(S_t, g)$$

All of the above updates can be done using any off-policy GVF algorithm, including those using clipping of IS ratios and gradient-based methods, and can include replay.

**Model Update without Importance Sampling**  Overloading notation, let us define the action-value variants $r_\gamma(s, a, g)$ and $\Gamma(s, a, g)$. We get similar updates to above, now redefining

$$\delta_t^r = R_{t+1} + \gamma_{g,t+1} r_\gamma(S_{t+1}, \pi_g(S_{t+1}), g) - r_\gamma(S_t, A_t, g)$$

and using update $\delta_t \nabla r_\gamma(S_t, A_t, g)$. For $\Gamma$ we have

$$\delta_t^\Gamma = m(S_{t+1}, g)\gamma_{t+1} + \gamma_{g,t+1}\Gamma(S_{t+1}, \pi_g(S_{t+1}), g) - \Gamma(S_t, A_t, g)$$

We then define $r_\gamma(s, g) \stackrel{\text{def}}{=} r_\gamma(s, \pi_g(s), g)$ and $\Gamma(s, g) \stackrel{\text{def}}{=} \Gamma(s, \pi_g(s), g)$ as deterministic functions of these learned functions.

**Restricting the Model Update to Relevant States**  Recall, however, that we need only query these models where $d(s, g) > 0$. We can focus our function approximation resources on those states. This idea has previously been introduced with an interest weighting for GVFs (Sutton et al., 2016), with connections made between interest and initiation sets (White, 2017). For a large state space with many subgoals, using goal-space planning significantly expands the models that need to be learned, especially if we learn one model per subgoal. Even if we learn a model that generalizes across subgoal vectors, we are requiring that model to know a lot: values from all states to all subgoals. It is likely such a models would be hard to learn, and constraining what we learn about with $d(s, g)$ is likely key for practical performance.

The modification to the update is simple: we simply do not update $r_\gamma(s, g), \Gamma(s, g)$ in states $s$ where $d(s, g) = 0$.[3] For the action-value variant, we do not update for state-action pairs $(s, a)$ where $d(s, g) = 0$ and $\pi_g(s) \neq a$. The model will only ever be queried in $(s, a)$ where $d(s, g) = 1$ and $\pi_g(s) = a$.

**Learning the relevance model $d$** We assume in this work that we simply have $d(s, g)$, but we can at least consider ways that we could learn it. One approach is to attempt to learn $\Gamma$ for each $g$, to determine which are pertinent. Those with $\Gamma(s, g)$ closer to zero can have $d(s, g) = 0$. In fact, such an approach was taken for discovering options (Khetarpal et al., 2020), where both options and such a relevance function are learned jointly. For us, they could also be learned jointly, where a larger set of goals start with $d(s, g) = 1$, then if $\Gamma(s, g)$ remains small, then these may be switched to $d(s, g) = 0$ and they will stop being learned in the model updates.

**Learning the subgoal-to-subgoal models** Finally, we need to extract the subgoal-to-subgoal models $\tilde{r}_\gamma, \tilde{\Gamma}$ from $r_\gamma, \Gamma$. The strategy involves updating towards the state-to-subgoal models, whenever a state corresponds to a subgoal. In other words, for a given $s$, if $m(s, g) = 1$, then for a given $g'$ (or iterating through all of them), we can update $\tilde{r}_\gamma$ using

$$(r_\gamma(s, g') - \tilde{r}_\gamma(g, g'))\nabla\tilde{r}_\gamma(g, g')$$

and update $\tilde{\Gamma}$ using

$$(\Gamma(s, g') - \tilde{\Gamma}(g, g'))\nabla\tilde{\Gamma}(g, g').$$

Note that these updates are not guaranteed to uniformly weight the states where $m(s, g) = 1$. Instead, the implicit weighting is based on sampling $s$, such as through which states are visited and in the replay buffer. We do not attempt to correct this skew, as mentioned in the main body, we presume that this bias is minimal. An important next step is to better understand if this lack of reweighting causes convergence issues, and how to modify the algorithm to account for a potentially changing state visitation.

## C.2 A General Algorithm for Learning Option Policies

Finally, we need to learn the option policies $\pi_g$. In the simplest case, it is enough to learn $\pi_g$ that makes $r_\gamma(s, g)$ maximal for every relevant $s$ (i.e., $d(s, g) > 0$). We can learn the action-value variant $r_\gamma(s, a, g)$ using a Q-learning update, and set $\pi_g(s) = \operatorname{argmax}_{a \in \mathcal{A}} r_\gamma(s, a, g)$, where we overloaded the definition of $r_\gamma$. We can then extract $r_\gamma(s, g) = \max_{a \in \mathcal{A}} r_\gamma(s, a, g)$, to use in all the above updates and in planning. In our own Pinball Experiment, this strategy is sufficient for learning $\pi_g$.

More generally, however, this approach may be ineffective because maximizing environment reward may be at odds with reaching the subgoal in a reasonable number of steps (or at all). For example, in environments where the reward is always positive, maximizing environment reward might encourage the option policy not to terminate.[4] However, we do want $\pi_g$ to reach $g$, while also obtaining the best return along the way to $g$. For example, if there is a lava pit along the way to a goal, even if going through the lava pit is the shortest path, we want the learned option to get to the goal by going around the lava pit. We therefore want to be reward-respecting, as introduced for reward-respecting subtasks (Sutton et al., 2022), but also ensure termination.

We can consider a spectrum of option policies that range from the policy that reaches the goal as fast as possible to one that focuses on environment reward. We can specify a new reward for

---

[3]More generally, we might consider using *emphatic weightings* (Sutton et al., 2016) that allow us to incorporate such interest weightings $d(s, g)$, without suffering from bootstrapping off of inaccurate values in states where $d(s, g) = 0$. Incorporating this algorithm would likely benefit the whole system, but we keep things simpler for now and stick with a typical TD update.

[4]It is not always the case that positive rewards result in option policies that do not terminate. If there is a large, positive reward at the subgoal in the environment, Even if all rewards are positive, if $\gamma_c < 1$ and there is larger positive reward at the subgoal than in other nearby states, then the return is higher when reaching this subgoal sooner, since that reward is not discounted as many steps. This outcome is less nuanced for negative reward. If the rewards are always negative, on the other hand, then the option policy will terminate, trying to find the path with the best (but still negative) return.

learning the option: $\tilde{R}_{t+1} = cR_{t+1} + (1-c)(-1)$. When $c = 0$, we have a cost-to-goal problem, where the learned option policy should find the shortest path to the goal, regardless of reward along the way. When $c = 1$, the option policy focuses on environment reward, but may not terminate in $g$. We can start by learning the option policy that takes the shortest path with $c = 0$, and the corresponding $r_\gamma(s, g), \Gamma(s, g)$. The constant $c$ can be increased until $\pi_g$ stops going to the goal, or until the discounted probability $\Gamma(s, g)$ drops below a specified threshold.

Even without a well-specified $c$, the values under the option policy can still be informative. For example, it might indicate that it is difficult or dangerous to attempt to reach a goal. For this work, we propose a simple default, where we fix $c = 0.5$. Adaptive approaches, such as the idea described above, are left to future work.

The resulting algorithm to learn $\pi_g$ involves learning a separate value function for these rewards. We can learn action-values (or a parameterized policy) using the above reward. For example, we can learn a policy with the Q-learning update to action-values $\tilde{q}$

$$\left( cR_{t+1} + c - 1 + \gamma_{g,t+1} \max_{a'} \tilde{q}(S_{t+1}, a', g) - \tilde{q}(S_t, A_t, g) \right) \nabla \tilde{q}(S_t, A_t, g)$$

Then we can set $\pi_g$ to be the greedy policy, $\pi_g(s) = \mathrm{argmax}_{a \in \mathcal{A}} \tilde{q}(s, a, g)$.

### C.3 Pseudocode putting it all together

We summarize the above updates in pseudocode, specifying explicit parameters and how they are updated. The algorithm is summarized in Algorithm 1, with a diagram in Figure 4. An online update is used for the action-values for the main policy, without replay. All background computation is used for model learning using a replay buffer and for planning with those models. The pseudocode assumes a small set of subgoals, and is for episodic problems. We provide extensions to other settings in Appendix C.4, including using a Double DQN update for the policy update. We also discuss in-depth differences to existing related ideas, including landmark states, UVFAs, and Goal-conditioned RL in Appendix D.

Note that we overload the definitions of the subgoal models. We learn action-value variants $r_\gamma(s, a, g; \theta^r)$, with parameters $\theta^r$, to avoid importance sampling corrections. We learn the option-policy using action-values $\tilde{q}(s, a; \theta^\pi)$ with parameters $\theta^\pi$, and so query the policy using $\pi_g(s; \theta^\pi) \stackrel{\text{def}}{=} \mathrm{argmax}_{a \in \mathcal{A}} \tilde{q}(s, a, g; \theta^\pi)$. The policy $\pi_g$ is not directly learned, but rather defined by $\tilde{q}$. Similarly, we do not directly learn $r_\gamma(s, g)$; instead, it is defined by $r_\gamma(s, a, g; \theta^r)$. Specifically, for model parameters $\theta = (\theta^r, \theta^\Gamma, \theta^\pi)$, we set $r_\gamma(s, g; \theta) \stackrel{\text{def}}{=} r_\gamma(s, \pi_g(s; \theta^\pi), g; \theta^r)$ and $\Gamma(s, g; \theta) \stackrel{\text{def}}{=} \Gamma(s, \pi_g(s; \theta^\pi), g; \theta^\Gamma)$. We query these derived functions in the pseudocode.

Finally, we assume access to a given set of subgoals. But there have been several natural ideas already proposed for option discovery, that nicely apply in our more constrained setting. One idea was to use subgoals that are often visited by the agent (Stolle & Precup, 2002). Such a simple idea is likely a reasonable starting point to make a GSP algorithm that learns everything from scratch, including subgoals. Other approaches have used bottleneck states (McGovern & Barto, 2001).

---

**Algorithm 1** Goal-Space Planning for Episodic Problems

---

Assume given subgoals $\mathcal{G}$ and relevance function $d$
Initialize table $v \in \mathbb{R}^{|\mathcal{G}|}$, main policy $\mathbf{w}$, model parameters $\theta = (\theta^r, \theta^\Gamma, \theta^\pi), \tilde{\theta} = (\tilde{\theta}^r, \tilde{\theta}^\Gamma)$
Sample initial state $s_0$ from the environment
**for** $t \in 0, 1, 2, ...$ **do**
    Take action $a_t$ using $q$ (e.g., $\epsilon$-greedy), observe $s_{t+1}, r_{t+1}, \gamma_{t+1}$
    `ModelUpdate`$(s_t, a_t, s_{t+1}, r_{t+1}, \gamma_{t+1})$
    `Planning()`
    `MainPolicyUpdate`$(s_t, a_t, s_{t+1}, r_{t+1}, \gamma_{t+1})$

---

---

**Algorithm 2** `MainPolicyUpdate`$(s, a, s', r, \gamma)$

---

$v_{\text{sub}} \leftarrow \max_{g \in \bar{\mathcal{G}}: d(s,g) > 0} r_\gamma(s, g; \theta) + \Gamma(s, g; \theta)\tilde{v}(g)$
$\delta \leftarrow r + \gamma\beta v_{\text{sub}} + \gamma(1 - \beta)\max_{a'} q(s', a'; \mathbf{w}) - q(s, a; \mathbf{w})$
$\mathbf{w} \leftarrow \mathbf{w} + \alpha\delta\nabla_{\mathbf{w}}q(s, a; \mathbf{w})$

---

 

**Algorithm 3** `Planning`$()$

---

**for** $n$ iterations, for each $g \in \mathcal{G}$ **do**
    $\tilde{v}(g) \leftarrow \max_{g' \in \bar{\mathcal{G}}: d(g,g') > 0} \tilde{r}_\gamma(g, g'; \tilde{\theta}^r) + \tilde{\Gamma}(g, g'; \tilde{\theta}^\Gamma)\tilde{v}(g')$

---

 

**Algorithm 4** `ModelUpdate`$(s, a, s', r, \gamma)$

---

Add new transition $(s, a, s', r, \gamma)$ to buffer $B$
**for** $g' \in \bar{\mathcal{G}}$, for multiple transitions $(s, a, r, s', \gamma)$ sampled from $B$ **do**
    $\gamma_{g'} \leftarrow \gamma(1 - m(s', g'))$
    // Update option policy
    $\delta^\pi \leftarrow \frac{1}{2}(r - 1) + \gamma_{g'}\max_{a' \in \mathcal{A}} \tilde{q}(s', a', g'; \theta^\pi) - q(s, a, g'; \theta^\pi)$
    $\theta^\pi \leftarrow \theta^\pi + \alpha^\pi \delta^\pi \nabla q(s, a, g'; \theta^\pi)$
    // Update reward model and discount model
    $a' \leftarrow \pi_{g'}(s'; \theta^\pi)$
    $\delta^r \leftarrow r + \gamma_{g'} r_\gamma(s', a', g'; \theta^r) - r_\gamma(s, a, g'; \theta^r)$
    $\delta^\Gamma \leftarrow m(s', g)\gamma + \gamma_{g'}\Gamma(s', a', g'; \theta^\Gamma) - \Gamma(s, a, g'; \theta^\Gamma)$
    $\theta^r \leftarrow \theta^r + \alpha^r \delta^r \nabla r_\gamma(s, a, g'; \theta^r)$
    $\theta^\Gamma \leftarrow \theta^\Gamma + \alpha^\Gamma \delta^\Gamma \nabla\Gamma(s, a, g'; \theta^\Gamma)$
    // Update goal-to-goal models using state-to-goal models
    **for** each $g$ such that $m(s, g) > 0$ **do**
        $\tilde{\theta}^r \leftarrow \tilde{\theta}^r + \tilde{\alpha}^r(r_\gamma(s, g'; \theta) - \tilde{r}_\gamma(g, g'; \tilde{\theta}^r))\nabla\tilde{r}_\gamma(g, g'; \tilde{\theta}^r)$
        $\tilde{\theta}^\Gamma \leftarrow \tilde{\theta}^\Gamma + \tilde{\alpha}^\Gamma(\Gamma(s, g'; \theta) - \tilde{\Gamma}(g, g'; \tilde{\theta}^r))\nabla\tilde{\Gamma}(g, g'; \tilde{\theta}^\Gamma)$

---

## C.4 EXTENDING GSP TO DEEP RL

It is simple to extend the above pseudocode for the main policy update and the option policy update to use Double DQN (Van Hasselt et al., 2016) updates with neural networks. The changes from the above pseudocode are 1) the use of a target network to stabilize learning with neural networks, 2) using polyak averaging to interpolate between the target network and the main network's weights, 3) changing the one-step bootstrap target to the DDQN equivalent, 4) adding a replay buffer for learning the main policy, and 5) changing the update from using a single sample to using a batch update. Because the number of subgoals is discrete, the equations for learning $\tilde{\theta}^r$ and $\tilde{\theta}^\Gamma$ does not change. We summarize these changes for learning the main policy in Algorithm 5 and for learning subgoal models in Algorithm 6.

---

**Algorithm 5** `MainPolicyDDQNUpdate`$(s, a, s', r, \gamma)$

---

Add experience $(s, a, s', r, \gamma)$ to replay buffer $D_{main}$
**for** $n_{main}$ mini-batches **do**
    Sample batch $B_{main} = \{(s, a, r, s', \gamma)\}$ from $D_{main}$
    $v_{\text{sub}}(s) = \max_{g \in \bar{\mathcal{G}}: d(s,g) > 0} r_\gamma(s, g; \theta) + \Gamma(s, g; \theta)\tilde{v}(g)$
    $Y(r, s', \gamma) = r + \gamma\beta v_{\text{sub}} + \gamma(1 - \beta)q(s', \max_{a'} q(s', a'; \mathbf{w}), \mathbf{w}_{target})$
    $L = \frac{1}{|B_{main}|}\sum_{(s,a,r,s',\gamma) \in B_{main}}(Y(r, s', \gamma) - q(s, a; \mathbf{w}))^2$
    $\mathbf{w} \leftarrow \mathbf{w} + \alpha\nabla_{\mathbf{w}}L$
    $\mathbf{w}_{target} \leftarrow \rho\mathbf{w} + (1 - \rho)\mathbf{w}_{target}$

---

---

**Algorithm 6** `ModelDDQNUpdate`$(s, a, s', r, \gamma)$

---

Add new transition $(s, a, s', r, \gamma)$ to buffer $D_{model}$

**for** $g' \in \bar{\mathcal{G}}$ **do**

    **for** $n_{model}$ mini-batches **do**

        Sample batch $B_{model} = \{(s, a, r, s', \gamma)\}$ from $D_{model}$

        $\gamma_{g'} \leftarrow \gamma(1 - m(s', g'))$

        // Update option policy

        $a' \leftarrow \mathrm{argmax}_{a' \in \mathcal{A}} \tilde{q}(s', a', g'; \theta^\pi)$

        $\delta^\pi(s, a, s', r, \gamma) \leftarrow \frac{1}{2}(r - 1) + \gamma_{g'} \tilde{q}(s', a', g'; \theta^\pi_{targ}) - q(s, a, g'; \theta^\pi)$

        $\theta^\pi \leftarrow \theta^\pi + \alpha^\pi \nabla_{\theta^\pi} \frac{1}{|B_{model}|} \sum_{(s,a,r,s',\gamma) \in B_{model}} (\delta^\pi(s, a, s', r, \gamma))^2$

        $\theta^\pi_{targ} \leftarrow \rho_{model} \theta^\pi + (1 - \rho_{model}) \theta^\pi_{targ}$

        // Update reward model and discount model

        $\delta^r(s, a, r, s', \gamma) \leftarrow r + \gamma_{g'}(\gamma, s') r_\gamma(s', a', g'; \theta^r{}_{targ}) - r_\gamma(s, a, g'; \theta^r)$

        $\delta^\Gamma(s, a, r, s', \gamma) \leftarrow m(s', g)\gamma + \gamma_{g'}(\gamma, s')\Gamma(s', a', g'; \theta^\Gamma{}_{targ}) - \Gamma(s, a, g'; \theta^\Gamma)$

        $\theta^r \leftarrow \theta^r - \alpha^r \nabla_{\theta^r} \frac{1}{|B_{model}|} \sum_{(s,a,r,s',\gamma) \in B_{model}} (\delta^r)^2$

        $\theta^\Gamma \leftarrow \theta^\Gamma - \alpha^\Gamma \nabla_{\theta^\Gamma} \frac{1}{|B_{model}|} \sum_{(s,a,r,s',\gamma) \in B_{model}} (\delta^\Gamma)^2$

        $\theta^r{}_{targ} \leftarrow \rho_{model} \theta^r + (1 - \rho_{model}) \theta^r{}_{targ}$

        $\theta^\Gamma{}_{targ} \leftarrow \rho_{model} \theta^\Gamma + (1 - \rho_{model}) \theta^\Gamma{}_{targ}$

    // Update goal-to-goal models using state-to-goal models

    . . . same as in prior pseudocode.

---

## C.5 Optimizations for GSP using Fixed Models

It is possible to reduce computation cost of GSP when learning with a fixed model. When the subgoal models are fixed, $v_{\text{sub}}$ for an experience sample does not change over time as all components that are used to calculate $v_{\text{sub}}$ are fixed. This means that the agent can calculate $v_{\text{sub}}$ when it first receives the experience sample and save it in the buffer, and use the same calculated $v_{\text{sub}}$ whenever this sample is used for updating the main policy. When doing so, $v_{\text{sub}}$ only needs to be calculated once per sample experienced, instead of with every update. This is beneficial when training neural networks, where each sample is often used multiple times to update network weights.

An additional optimization possible on top of caching of $v_{\text{sub}}$ in the replay buffer is that we can batch the calculation of $v_{\text{sub}}$ for multiple samples together, which can be more efficient than calculating $v_{\text{sub}}$ for a single sample every step. To do this, we create an intermediate buffer that stores up to some number of samples. When the agent experiences a transition, it adds the sample to this intermediate buffer rather than the main buffer. When this buffer is full, the agent calculates $v_{\text{sub}}$ for all samples in this buffer at once and adds the samples alongside $v_{\text{sub}}$ to the main buffer. This intermediate buffer is then emptied and added to again every step. We set the maximum size for the intermediate buffer to 1024 in our experiments.

## D Connections to UVFAs and Goal-Conditioned RL

There is a large and growing literature on goal-conditioned RL (GCRL). This is a problem setting where the aim is to learn a policy $\pi(a|s, g)$ that can be (zero-shot) conditioned on different possible goals. The agent learns for a given set of goals, with the assumption that at the start of each episode the goal state is explicitly given to the agent. After this training phase, the policy should generalize to previously unseen goals. Naturally, this idea has particularly been applied to navigation, having the agent learn to navigate to different states (goals) in the environment. Many GCRL approaches leverage UVFAs (Schaul et al., 2015).

This setting bears a strong resemblance to what we do in this work, but is notably different. Our models can be seen as goal-conditioned models—part of the solution—for planning in the general RL setting. GCRL, on the other hand, is a problem setting. Many approaches do not consider planning, but instead focus on effectively learning the goal-conditioned value functions or policies.

There is more work, however, using landmark states and planning, for GCRL. In addition to the goal given for GCRL, the landmark states can be treated as interim subgoals and UVFA models learned for these as well (Huang et al., 2019). Planning is done between landmarks, using graph-based search. The policy is set to reach the nearest goal (using action-values with cost-to-goal rewards of -1 per step) and learned distance functions between states and goals and between goals. These models are like our reward and discount models, but tailored to navigation and distances.

The idea of learning models that immediately apply to new subtasks, using successor features, is like GCRL but goes beyond navigation. The option keyboard involves encoding options (or policies) as vectors that describe the corresponding (pseudo) reward (Barreto et al., 2019). This work has been expanded more recently, using successor features (Barreto et al., 2020). New policies can then be easily obtained for new reward functions, by linearly combining the (basis) vectors for the already learned options. No planning is involved in this work, beyond a one-step decision-time choice amongst options.

# E    ADDITIONAL DETAILS ON LEARNING SUBGOAL MODELS

This section describes implementation details for learning subgoal models in the PinBall environment and errors observed in the learned models.

To ensure that we provide sufficient variety of data to learn the model accurately, when learning the subgoal models, the agent is randomly initialized in the environment at a valid state, ran in the environment for 20 steps with a random policy, then randomly reset again. To ensure that the agent gets sufficient experience near goal states, we initialize the agent, with a 0.01 probability, at states where $m(s, g) = 1$ for any $g$ with added jitter sampled from $U(-0.01, 0.01)$ for each feature. The model is trained for 300k steps in this data gathering regime.

We restrict model update to relevant states in our experiments. Because the only relevant experience for learning $r_\gamma$ and $\Gamma$ are samples where $d(s, g) > 0$, we maintain a separate buffer for each subgoal $g$ for learning $r_\gamma(s, g)$ and $\Gamma(s, g)$ such that all experience within that buffer are relevant. We require 10k samples in the buffer of each subgoal before learning for the corresponding $r_\gamma$ and $\Gamma$ begins, so that mini-batches are always drawn from a sufficiently diverse set of samples.

Similarly, a sample is only relevant for updating $\tilde{\Gamma}$ and $\tilde{r}_\gamma$ if $m(s, g) > 0$ for some $g$, but this might not be true for samples stored in the buffers for learning $\Gamma$ and $r_\gamma$. To be able to obtain a batch of samples where all samples are relevant for learning $\tilde{\Gamma}$ and $\tilde{r}_\gamma$, the agent uses another buffer that exclusively stores samples where $m(s, g) > 0$ to learn $\tilde{\Gamma}$ and $\tilde{r}_\gamma$.

We mentioned in Appendix C.1 that we take the simple approach to restricting model updates to states where $d(s, g) = 1$. However, this means an update could bootstrap off inaccurate estimates when learning from a sample $(s, a, r, s')$ if $d(s, g) > 0$ but $d(s', g) = 0$. In PinBall, this occurs when the agent starts within the relevance area for a subgoal but taking an action moves the agent outside of it. We attempt to alleviate this issue in practice by changing the estimation target for those state-action pairs to be the minimum possible target in the environment. Because we co-learn the option policy with $r_\gamma(s, a, g)$, we set this minimum value to $\frac{1}{1-\gamma} r_{min}$. If the network can learn this target well, then the learned option policy will not leave the relevance area.

We also address the issue that for some fixed $d$, it is possible that not all states where $d(s, g) > 0$ could reach the subgoal. This can negatively affect the quality of $v_{\text{sub}}$ as our algorithm assumes that goal $g$ is reachable from state $s$ via the option policy if $d(s, g) > 0$. While this source of error did not seem to affect GSP in our experiments, it might be important in other environments, so we describe the modification to address this problem here. From these states, the agent should not consider these subgoals when doing background planning ($g'$ is not reachable from $g$ despite $\tilde{d}(g, g') = 1$) and projection ($g'$ is not reachable from $s$ despite $d(s, g') = 1$). We check for these states by seeing if the learned $\Gamma(s, g)$ is near 0, which indicates that it is either very difficult or impossible to reach $g$ from $s$. For states with $\Gamma(s, g)$ very near 0, we can set $d(s, g) = 0$ for the purpose of background planning and projection, but not for learning $\Gamma(s, g)$ as it might be initialized to a low value. In our experiments, we set this threshold to 0.

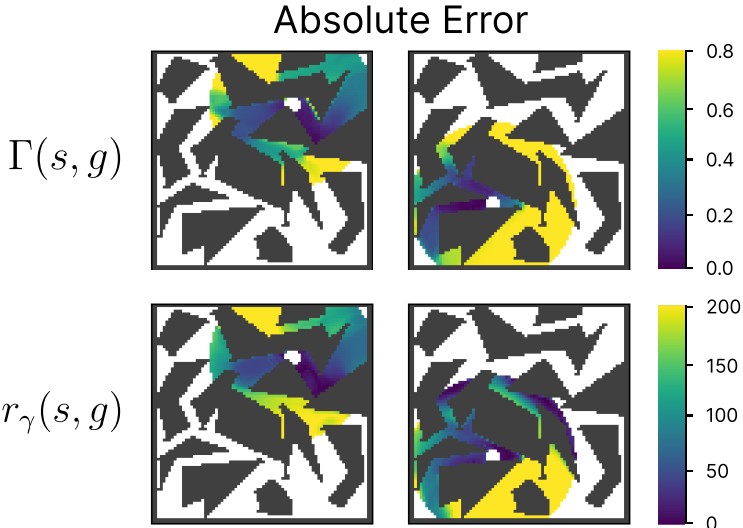

Figure 11: A heatmap of the absolute error of $\Gamma$ and $r_\gamma$ for two different subgoal models learned at various $(x, y)$. While the absolute error from states near subgoals can be quite low, they increase substantially as the state gets further away. White indicates that $d(s, g) = 0$.

### E.1 Error of Learned Subgoal Models

To better understand the accuracy of our learned subgoal models, we performed roll-outs of the learned option policy at different (x,y) locations (with 0 velocity) across the environment and compared the true $r_\gamma$ and $\Gamma$ with the estimated values. Figure 11 shows a heatmap of the absolute error of the model compared to the ground truth, with the mapping of colors on the right. The models learned tend to be more accurate closer to the goal, and less accurate further away. The absolute error of $\Gamma$ can be as low as 0.01 close to the goal, but increase to 0.2 and higher further away. Similarly, the absolute error for $r_\gamma$ can be as low as below 10 near goals, but can increase over 100 further away. While the magnitudes of errors are not unreasonable, they are also not very near zero. This results is encouraging in that inaccuracies in the model do not prevent useful planning.

## F Additional Experiment Details

This section provides additional details for the PinBall environment, the various hyperparameters used for DDQN and GSP, and the hyperparameters sweeps performed. The experiments described in the main body, along with the hyperparameter sweeps, used approximately 10.7 CPU days on an Apple M1 chip.

The pinball configuration that we used is based on the "slightly harder configuration" found at `http://irl.cs.brown.edu/pinball/`. The Python implementation of PinBall was taken from `https://github.com/amarack/python-rl`, which was released under the GPL-3.0 license. We have modified the environment to support additional features such as changing terminations, visualizing subgoals, and various bug fixes.

**Network Architecture** We used neural networks for learning the main policy, $\Gamma$, and $r_\gamma$. For experiment 1, we used a neural network with hidden layers $[256, 256, 128, 128, 64, 64]$ for the main policy and $[256, 256, 128, 128, 64, 64, 32, 32]$ for $\Gamma$ and $r_\gamma$. For experiment 2, we used a neural network with hidden layers $[128, 128, 64, 64]$ for the main policy and $[128, 128, 128, 128, 64, 64]$ for $\Gamma$ and $r_\gamma$. We used ReLU activation function for each layer aside from the output layer. The network's bias weights are initialized to 0.001 and other weights were initialized using He uniform initialization (He et al., 2015). Each network output a vector of length 5, one for each action.

### F.1 EXPERIMENT HYPERPARAMETERS

For both experiments, we used the Adam optimizer for training both the main policy and the subgoal models. We used the default hyperparameters for Adam except the step-size ($b_1 = 0.9, b_2 = 0.999, \epsilon = 1e^{-8}$). The main policy was trained with 4 mini-batches per step with batch size of 16, while the subgoal models were trained with 1 mini-batch per step with the same batch size. We used $\epsilon$-greedy exploration strategy, with $\epsilon$ fixed to $\epsilon = 0.1$ in our experiments.

For experiment 1, $\gamma = 0.99$, $\alpha^\pi = \alpha^r = \alpha^\Gamma = 5e^{-4}$, and $\rho_{model} = 0.4$. For experiment 2, $\gamma = 0.95$, $\alpha^\pi = \alpha^r = \alpha^\Gamma = 1e^{-3}$, and $\rho_{model} = 0.1$. We selected the learning rate for Adam and the polyak averaging rate $\rho$ for updating the main policy in each experiment using the methodology described in the section below.

### F.2 HYPERPARAMETER SWEEP METHODOLOGY

For experiment 1, we swept the baseline DDQN algorithm for polyak averaging rate $\rho \in [0.0125, 0.025, 0.05, 0.1]$ and learning rate in $[1e^{-3}, 5e^{-4}, 3e^{-4}, 1e^{-4}]$ across 4 seeds. We found that $\rho = 0.025$ and learning rate of $5e^{-4}$ had the highest average reward rate in our sweep and used them when running both DDQN and GSP across seeds in the experiment.

For experiment 2, we swept DDQN for polyak averaging rate $\rho \in [1.0, 0.8, 0.4, 0.2, 0.1, 0.05, 0.025]$ and learning rate in $[1e^{-2}, 5e^{-3}, 1e^{-3}, 5e^{-4}]$ for 8 seeds. We find that $\rho = 0.05$ and learning rate of $1e^{-3}$ had the highest average reward rate out of all configurations swept and used these hyperparameters for all DDQN runs in the experiment. For GSP, we used $\rho = 0.8$ and learning rate of $1e^{-3}$.

## G COMPARING GSP TO OTHER DYNA ALTERNATIVES

In this section, we compare GSP against other basic background planning algorithms. Namely, we compare against DDQNx2, a DDQN agent that is given double the amount of computational budget per step compared to our baseline algorithm, and Dyna with options (Dyno), a natural alternative to use option model for background planning.

As mentioned in Section 4, DDQN can be viewed as a background planning algorithm when the replay buffer is viewed as a non-parametric model. Providing DDQN with double the number of mini-batch updates attempts to answer the question of what if GSP's background planning resources was instead dedicated to additional one-step updates. Note that in this experiment, our DDQNx2 implementation took 50% additional wall-clock time to run when compared to our GSP implementation.

Dyna with options (Dyno) is a basic algorithm that incorporates option models into Dyna such that the agent learns about both action values and option values $Q : \to S \times A \cup O$. Dyno's behaviour policy then includes both actions and options. If an option $\pi_j$ is selected when taking a greedy action according to $Q$, then the first action given by $\pi_j$ is executed. The model in Dyna needs to include option models, which allows the agent to reason about accumulated rewards under an option, and outcome states after executing an option. Otherwise, the framework is identical to Dyna. It is a simple, elegant extension on Dyna that allows for planning with temporal abstraction. However, this approach has several limitations. One limitation is that as we include new options—more abstraction—our value function needs to reason over more actions. Our proposed algorithm, GSP, allows the agent to obtain the benefits of abstraction, without modifying the form of the policy. Another limitation is that the model in Dyna is the standard state-to-state model. Though Dyna with options has not been extended to function approximation — somewhat surprisingly — the natural extension suffers from similar problems of model errors and the use of expectation models as standard Dyna.

We compare DDQNx2, Dyno, and GSP in the simple PinBall environment. For Dyno, we use the same subgoal-conditioned models pre-trained for GSP as options models and set the predicted next state of each option to $(x_g, y_g, 0, 0)$. We found Dyna with options difficult to get working. Instead, we used a modified version that only plans over options. This avoids learning and using primitive action models. We see in Figure 12 that this modified variant actually outperformed DDQN initially, but leveled off at a suboptimal level of performance and overall learned slower than GSP. We also

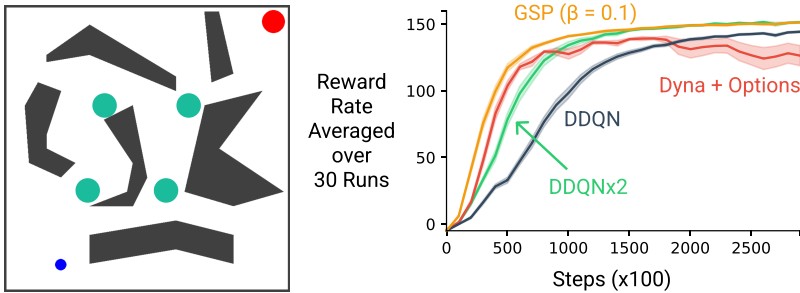

Figure 12: **(left)** The simple PinBall environment. **(right)** The performance of Dyna with options (Dyno), DDQNx2, DDQN, and GSP ($\beta = 0.1$) in the simple PinBall environment. Dyno learned slower than GSP and converged to a lower performance point when compared to GSP. DDQNx2, despite requiring an additional 50% wall-clock time when compared to GSP, learned at a slower rate but converged to the same optimal peformance.

find that DDQNx2 performed better than DDQN, but was unable to perform better than GSP despite requiring more wall-clock time, highlighting GSP's computational efficiency.

## H    EXPERIMENTS WITH LUNAR LANDER

To see how GSP can be applied to other problems, we ran GSP in the Lunar Lander environment (Brockman et al., 2016). The environment specification follow OpenAI Gym's `LunarLaner-v2` environment. We provide the agent with 9 subgoals, with one terminal subgoal when the agent lands safely on the landing pad and the rest of the subgoals laid throughout the environment at different (x,y) locations in an arrow-like fashionx. As the (x,y) coordinates are continuous, we take a similar approach of defining a small region around each coordinate to for subgoal termination, and define a larger initiation area around each subgoal. We show the non-terminal subgoals in Figure 13. We compare GSP, DDQN, DDQNx4 (DDQN with 4x the amount of planning steps), and approximate LAVI. We evaluated GSP with $\beta = 0.01$ as we found it to be the best performing $\beta \in [0.001, 0.01, 0.05]$ in our experiments.

We see in Figure 13 that GSP outperforms DDQN, DDQNx4, and approximate LAVI. Overall, we found that subgoal-conditioned models are more difficult to learn in Lunar Lander, with the learned reward models and discount model having an average absolute error of around 5 and 0.1 respectively from 200 monte-carlo rollouts of the policy in the environment. This aligns with the poor performance of approximate LAVI and the lower value of $\beta$ that was found to be good for GSP. Surprisingly, DDQNx4 performed worse than GSP and DDQN despite performing 4 times the number of batch updates that DDQN performs per step. We hypothesize that this is because the increased number of updates causes the agent to fit to a suboptimal solution based on insufficient data, thus making the rate of improvement slower.

## I    INVESTIGATING GSP WITH DIFFERENT BETA

In subgoal-value bootstrapping (Equation 4), the hyperparameter $\beta$ represents the tradeoff between fully using the quickly updated but approximate subgoal values $v_{\text{sub}}(s)$ and the standard bootstrap target. We investigate the impact of $\beta$ in the harder pinball environment shown in Figure 5. We ran GSP with $\beta \in [0.0, 1e^{-3}, 0.1, 0.5, 1.0]$. Note that $\beta = 0.0$ is equivalent to DDQN, and $\beta = 1.0$ is equivalent to approximate LAVI. We see in Figure 14 that with $\beta = 0.5$ and $\beta = 1.0$, GSP gets similar fast initial learning, but converges to a lower final performance. For $\beta = 1e^{-3}$ very close to 0, we see that performance is more like DDQN. But even for such a small $\beta$ we get improvements.

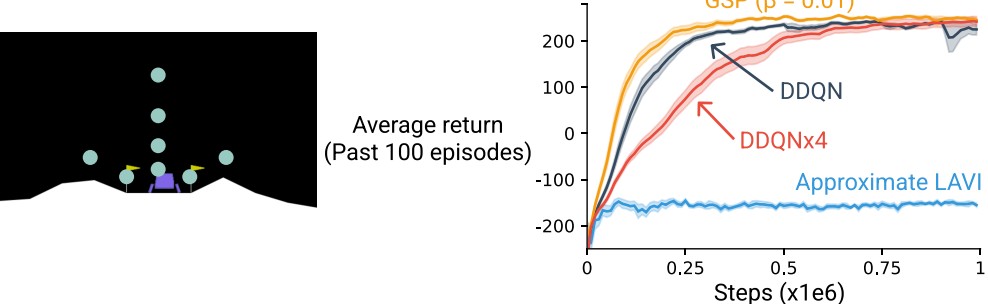

Figure 13: **(left)** The locations of non-terminal subgoals in Lunar Lander, shown with light green circles. **(right)** Mean return over 30 independent runs of the past 100 epsides of GSP, DDQN, DDQNx4, and approximate LAVI in Lunar Lander. GSP learned more quickly than DDQN and DDQNx4 and converged to the same level of performance. The performance of approximate LAVI quickly plateaued as it was unable to overcome the suboptimality of the learned subgoal models.

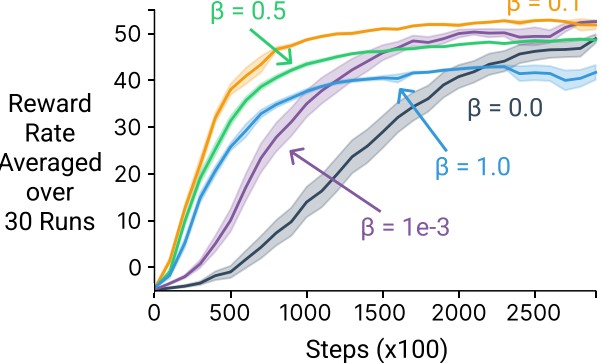

Figure 14: Performance in the harder PinBall environment for GSP with a variety of $\beta$, with the standard error shown. Even just increasing to $\beta$ from 0 to 0.1 allows GSP makes it learn much faster than DDQN by leveraging learned subgoal values. Once $\beta$ is at 1, where it fully bootstraps off of potentially suboptimal subgoal values, GSP still learns quickly but arrives at a suboptimal policy.

