# OpenReview forum: "Goal-Space Planning with Subgoal Models"
_ICLR.cc/2023/Conference — Submitted to ICLR 2023_

### Official Review · Reviewer_bsyJ · 2022-10-14

**Confidence:** 3
**Correctness:** 4
**Technical Novelty And Significance:** 2
**Empirical Novelty And Significance:** 3
**Recommendation:** 5

**Clarity, Quality, Novelty And Reproducibility:**


* Clarity: The paper is well written.
* Quality: The paper is technically well written, although I have the feeling it could be more condensed (you write quite extensive paragraphs).
* Novelty: Low to moderate (see my comments above).
* Reproducibility: I do not see any mention of public code, but I do think the method could be reimplemented (up to some details, see my comments above).

**Details Of Ethics Concerns:**

-

**Strength And Weaknesses:**

Strenghts:

* The paper is well written.
* The topic (planning over landmarks) is relevant.
* The paper is technically sound.

Weaknesses:

* The contribution is a bit incremental. The authors essentially 1) prespecify a set of subgoals, 2) prespecify the relation between subgoals and the underlying state space, and 3) pretrain a set of models to reach between these subgoals. The pretrained value function on the subgoals is then used as part of the bootstrap target in a downstream RL agent. This agent manages to outperform the DDQN baseline, but this is not too surprising given the extra amount of prior knowledge and pretraining given to the agent. I think the comparison would be more fair if the agent at least had to learn the value function to move between subgoals on the fly (during training).
* Experimental evidence does not include many environments. The main paper only contains one task: PinBall, of which two instantiations are tested. The appendix does contain a second environment (LunarLandar), which I would definitely move to the main paper.

Smaller weaknesses:
* In your nonstationarity experiments, you indicate that “when the world changes, the agent recognizes that it has changed, and resets all counts”. I do not get how this works. Do you mean you hardcoded this? That does not seem to be fair, as the original Dyna+ implementation did not do that either?
* You say you “somewhat randomly selected subgoals across the PinBall environment”. What does this mean? Looking at the pictures, they seem to be rather evenly distributed, which seems crucial for your method to work well. I think you handpicked them with a little added noise right, which is fine, but then describe the actual procedure you followed.
* p4: I think you never introduce d(s,g)? I therefore have trouble understanding what a ‘local model’ really means.


**Summary Of The Paper:**

This paper studies fast background planning, based on an abstract model over subgoals. The subgoal space is predefined, and the authors then learn expected return and discount models between subgoals, which are used in a tabular dynamic programming procedure. The resulting value function is added to the bootstrapping of the lower-level RL agent to speed up learning. Empirical results on the Pinball environment show that learning is indeed faster.

**Summary Of The Review:**

This paper has its pros and cons. I think it is technically sound and well written, and the method works. However, my biggest issue is that the research question is a bit incremental, and the results therefore not too surprising (the method has much prior knowledge over the DDQN baseline). I am also a bit worried about the experimental evaluation on the PinBall environment only. I will therefore give a borderline vote.

---

### Official Review · Reviewer_1SkJ · 2022-10-24

**Confidence:** 4
**Correctness:** 4
**Technical Novelty And Significance:** 2
**Empirical Novelty And Significance:** 2
**Recommendation:** 5

**Clarity, Quality, Novelty And Reproducibility:**

[clarity] The organizational structure of the paper should be improved. It was not entirely clear until Eq. (4) that the Q-function was a single model-free policy that was being trained. To address this issue, redoing Fig. 1 will help considerably; readers with a mindset of using subgoals as part of a model-based planning framework may approach the figure and see an illustration of a roadmap-like planning framework. However, the figure illustrates the *training procedure* that uses approximate updates from the network of subgoals to more quickly train the model-free policy. [We note that this detail was alluded to in the introduction alongside the discussion of Dyna, but it was not made explicit that this was the approach taken by the proposed method.]
Relatedly (a suggestion for improving clarity), the "putting it all together" Sec. 3.4 comes far too late, and I recommend that the authors consider how to make it clear earlier what learned functions are necessary and how they relate to one another. This way, the remaining subsections in Sec. 3 are focused more on the low-level technical details rather than a mix of critical high-level concepts and low-level details. As of now, the paper is fairly hard to follow.

[clarity] (I believe this to be accurate, but I am uncertain) It should be made clearer that "Approximate LAVI" is GSP with beta=1 and DDQN is GSP with beta=0. It was not clear that this was the relationship between the proposed approach and the baselines. Making this detail very clear in both the text and the figures would help understanding.

[novelty] The approach is novel, though (discussed in some detail above), the significance is perhaps quite low. It is somewhat difficult to tell how effective this approach will be in general, since the experiments are so limited in scope.

[Very small note; no impact on review] On one of my devices (an iPad) the right side of Fig. 13 did not render and appeared as if it was missing. It rendered properly on another device (an Apple laptop), so it is unclear why this occurred. Perhaps see if there is anything that can be done. I find that embedding high-resolution PNG images helps ensure portability with little drop in quality or increase in image size.

**Strength And Weaknesses:**

- [strength] The approach is novel and delivers on the promised improvements to learning speed for the given experimental setup. The theoretical discussion is also self-contained and fairly comprehensive and the appendix contains considerable detail.
- [weakness] The choice of beta is a critical piece of the proposed work, as it trades off between targeting the (potentially suboptimal) value from the subgoal model and the otherwise slow-to-converge q-value. In general, the main body of the paper is somewhat lacking on results that show how this value was chosen. Appendix I has some results that show (for an experiment not included in the main body of the paper) how changing the value of beta impacts the rate of learning and asymptotic performance. I believe that the two baselines shown (DDQN and approximate LAVI) correspond to extreme values of beta, yet clarifying this point and including a more detailed analysis in the main body of the paper (not the abstract) would be useful.
- [weakness] The paper is missing a discussion of LAVI and the novelty of the proposed approach with respect to LAVI. Since the paper is contrasting to an "approximate LAVI" baseline and LAVI seems the work most similar to the proposed method, a statement summarizing its contributions and also the novelty of the proposed work with respect to LAVI is an important detail.
- [weakness] Critically, the experiments shown in the main body of the paper are quite limited, and exist in only a single relatively-simple Pinball environment. While there is some discussion in the appendix of results in the popular Lunar Lander domain, the impact of the proposed approach is somewhat limited. Regardless, seeing results on a different domain (e.g., the Lunar Lander) in the *main body* of the text is helpful for understanding how this work can be used in other domains; I believe that this results should be moved to be in the main text. Relatedly, there are some questions about the additional results included in the appendix that could use further scrutiny. In particular the DDQNx# results are a bit odd. The explanation of the poor performing DDQNx4 results for the Lunar Lander domain is reasonable, though the improvements for the DDQNx2 results in the simple pinball domain point to lack of computational effort as a potential source of performance limitations of the DDQN approach and improvements to the amount of computation is not the claimed central advantage of the approach; further discussion of this point would be helpful and additional analysis or prose to clarify the trade off between performance, computation, and learning rate would be a welcome addition to the paper (even if it only appears in the appendix).
- [weakness] There is little to no discussion about how the subgoals should be chosen. The authors acknowledge that *learning* these subgoals is beyond the scope of the paper, yet the omission of a clear procedure to obtain these subgoals or a discussion of what makes for a "well-specified" set of subgoals is a notable omission from the paper. The paper would benefit from at least some cursory evaluation of how the results would change if the subgoals were changed; this is an important practical consideration for someone who may wish to build upon this approach.
- [unsure, but clarity would be helpful] Could the authors clarify on what happens if the subgoals are chosen poorly? The motivation for Eq. (4) from Eq. (3) suggests that beta exists primarily to compensate for not-yet-converged values of v_sub and also allow for optimal plans that do not follow the subgoals. For the latter problem, would a `max` operator not be more suitable, in which the optimization objective would select max( q(s_t+1, a), v_sub ). It could avoid the need to tune the value of beta and if the subgoals were poorly chosen, it would avoid entirely a reliance on those values during optimization.

**Summary Of The Paper:**

This paper proposes "goal-space planning", an approach to training a model-free policy that uses user-specified subgoal states to perform approximate longer-horizon bellman updates and thus accelerate learning. The central novelty of the approach is the "subgoal-value bootstrapping" update that balances updating the Q-function via a standard TD error and a TD error borne of values associated with navigation to the nearest subgoal and then motion between those subgoals until the goal can be reached. Results are shown on two variations of the "Pinball" environment originally proposed for option/skill-learning and in both the authors show improvements in convergence rate and competitive long-term performance; additional results appear in the appendix, including some on the Lunar Lander domain.

**Summary Of The Review:**

Overall, the proposed approach is interesting and shows some limited improvements in the scenarios in which a user can fairly easily specify some well-chosen subgoals. I believe that the paper's main weaknesses stem from a lack of clarity and somewhat poor organization. The language in the main body of the paper can be streamlined to communicate the same message in fewer words, which will improve clarity. This additional space can be used to incorporate some of the important experiments that should not have been relegated to the appendix (see above). The paper should aim to make those changes and ensure that the discussion of the relocated experiments addresses my concerns about the general utility and performance of the approach.

---

### Official Review · Reviewer_sJBg · 2022-10-24

**Confidence:** 4
**Correctness:** 3
**Technical Novelty And Significance:** 3
**Empirical Novelty And Significance:** 3
**Recommendation:** 5

**Clarity, Quality, Novelty And Reproducibility:**

In terms of clarity, while most of the paper is well written there were some important points of confusion. One of the central ones was the way the paper uses the term model.  Commonly the term model is used in RL settings to refer to the MDP as a whole, but here it seems to just mean any function.

I was also a bit surprised to see that the paper did not contain a related works section. While the authors try to cover some related work in different sections, particularly in introduction, it would still be useful to have a separate section. First off it would make it easier for readers to position the work within the larger literature, as there is now one section they can go to without worrying about reading through the entire paper and see the various connections. Secondly it would give the authors also a chance to go deeper into the various related work without worrying about breaking the flow of the current section. Also there is a lot of work related to using symbolic models as a way to identify and plan at an abstract level. I didn’t see any mention or discussion with respect to such works. Some useful starting points at looking at works in this direction might be [3], and [4].

In terms of novelty, while the use of subgoals and temporal abstractions are popular within RL, the formulation itself is novel. Also, the initial results seem promising, I do think the current set of experiments doesn’t clearly validate the claims being made. As for the reproducibility, the appendix does include a lot of the implementation details, however the authors did not provide the code used for the evaluation.

[3] Yang, Fangkai, et al. "Peorl: Integrating symbolic planning and hierarchical reinforcement learning for robust decision-making." arXiv preprint arXiv:1804.07779 (2018).

[4] Illanes, León, et al. "Symbolic plans as high-level instructions for reinforcement learning." Proceedings of the international conference on automated planning and scheduling. Vol. 30. 2020.


**Strength And Weaknesses:**

First off, to the best of my knowledge the methods discussed in the paper are unique and touch upon many issues usually discussed in relation to planning with learned models. The closest related work I know of are in the realm of state-aggregation based abstraction scenarios where similar bi-level planning methods have been discussed(cf. Aggregation chapter in [1] or more generally check [2] for state abstraction). Connected ideas have also been studied in the context of temporal abstraction in relation to hierarchical planning (more generally or more specifically as in the context of task and motion planning), but their application in the context of RL I believe is novel. The results also seem to indicate that the method does provide some advantage over some of the current approaches.

To me the biggest issue with the proposed approach is the assumption that the subgoals are given. Currently the main paper simply includes a line that identification of the subgoals and initiation set are part of the subgoal discovery process. However, I didn’t see any discussion on what form the process takes. I am assuming the subgoal discovery process can’t be trivial. After all, in long horizon sparse reward problem settings, you have to discover enough subgoals that it could lead you to the eventual high reward states. Similarly, applying this in the context of non-stationarity, aren’t you assuming that the subgoals already cover state spaces relevant to the change.

Another point that was unclear to me was about the use of beta itself. If the purpose of not relying simply on the v_sub is to avoid following suboptimal policy, shouldn’t the value of beta be reduced over time. Because if your target value still includes a component that comes from v_sub, couldn’t that result in suboptimal policies?

In terms of evaluation, while the paper does present a few different experiments, I don’t believe there are any results that clearly shows that the current method provides a clear advantage over baseline when both systems are starting from scratch for standard setting. Experiment 1 is the result that comes closest to evaluating this, however the authors chose not to include the effort needed for model learning in that evaluation.  I understand that the experiment is framed in terms of a pre-learned model, but that has not been the focus of the paper. Throughout the paper, the primary focus was on a standard RL problem where the system starts without any knowledge of the environment and identifies a policy. Also given the fact that the methods were compared in terms of number of steps, this would have been an easy comparison to make.

[1] Bertsekas, Dimitri. Reinforcement learning and optimal control. Athena Scientific, 2019.

[2] Li, Lihong, Thomas J. Walsh, and Michael L. Littman. "Towards a Unified Theory of State Abstraction for MDPs." AI&M. 2006.

**Summary Of The Paper:**

The paper looks at how one could improve the performance of RL agents by leveraging planning at the level of subgoals. The authors argue that by restricting planning at a higher level of abstraction the method side-steps many of the problems associated with traditional model-based planning approaches, like inaccuracy of the learned model. Specifically they construct a goal-space MDP that reflects transitions in the terms of subgoals and use value functions for this abstract MDP to bootstrap the learning of the agent operating at the more concrete level.

**Summary Of The Review:**

While I think the idea and the direction is interesting, the current version of the paper currently falls in the borderline region of acceptance for me. As mentioned, I have some concerns about the soundness of some of the assumptions and methods. I also feel the experiments could also be improved.

---

### Official Review · Reviewer_XCiE · 2022-10-26

**Confidence:** 4
**Correctness:** 3
**Technical Novelty And Significance:** 2
**Empirical Novelty And Significance:** 2
**Recommendation:** 3

**Clarity, Quality, Novelty And Reproducibility:**

The paper is clearly written, and it would be straight forward to implement the methods as explained.
As for novelty, the idea of using state abstractions and subgoals to improve planning efficiency is not particularly novel, specially in the case where it is assumed that the subgoals are given.

**Strength And Weaknesses:**

Strengths:
- The paper proposes  an interesting approach to incorporate subgoals value estimates into Q-learning updates.
- The paper was clearly written and it is easy to follow.



Weakness:
- The idea is not very novel, the proposed approach seems to be very similar to HPA* ( https://webdocs.cs.ualberta.ca/~mmueller/ps/hpastar.pdf ), where the planning approach is to create an abstraction of the original state space to plan quickly at the abstract level. Granted, this was done in the context of classical planning and path finding, but the main concept is similar.

- I understand the paper states that the focus is to plan when subgoals are given, so how to identify the subgoals is not covered, but this is a crucial step for abstract planning to work. Where do those abstractions come from? Not having explored that question, limits the contribution of the paper quite a bit.


- The results do not appear to be significant. Figure 5 shows that GSP achieves an average reward of approx. 50 and DDQN of approx 40. Based on the description of the problem, the agent received a reward of 10,000 when reaching the goal, and -5 for every time-step. This means that a return of 50 takes (10,000 - 50) / 5 = 1,990 steps, while a return of 40 takes (10,000 - 40) / 5 = 1,992 steps.
In fact, every increase in reward of 5 points is a reduction of 1 step on average, so at training step 1000  where there seems to be a large difference between GSP and DDQN (40 vs. 10), the difference in the number of steps to reach the goal is 6.



**Summary Of The Paper:**

The paper proposes using a state abstractions defined as subgoals to improve efficiency by doing planning on the background while learning action-value estimates.
Given a set of subgoals, which is a many-to-one mapping of states to abstract states, the proposed method runs dynamic programming to plan in the abstract space. The current estimate of the subgoal value function is then included in the standard Q-learning update used in the original state space.
The results compare the proposed method to DDQN and Approx Lavi on the classic PinBall environment.

**Summary Of The Review:**

An interesting paper tackling the use of state abstractions to improve planning efficiency.
Unfortunately, the paper falls short in standing out in novelty, and the improvements shown in the experiment do not appear to be very significant.
I think there is potential for this line of work, but in its current state there's quite a bit of room for improvement.

---

### Author Response · Authors · 2022-11-15
**Response to Reviews**

We appreciate that the reviewers either felt (a) that this idea is not novel and has been done before (in classical planning) and/or (b) avoids the one key problem which is subgoal discovery and/or (c) does not demonstrate wins on interesting enough environments. This is quite the mountain of concerns to overcome. We believe we are trying to do something inherently difficult with this work: consider a new planning framework, that builds on many previous ideas but has several key but subtle differences, and test it carefully in small controlled experiments. **Let us try to address these concerns below:**

(a) **Regarding its relationship to prior work, goal-space planning does indeed use ideas from different parts of planning**. There are aspects of Dyna, hierarchical planning, options and landmark value iteration in this work. We are not the first to recognize the utility of planning in an abstract space, nor the first to recognize the utility of options for temporal abstraction in planning. **However, we argue that goal-space planning is a novel and an importantly different way to leverage these ideas.** GSP uses **abstract planning with imperfect models to guide low-level decisions. Importantly, GSP does not take away the ability for the agent to adapt its low-level decisions using direct experience, allowing it to overcome imperfections in planning abstractions and learned models.** This structure augments model-free algorithms, and views the role of planning in a specific manner: to improve sample efficiency and reactivity to environment changes, under computational restrictions.

**Regarding (b) and (c), the approach of this paper is to motivate why GSP should be better than current approaches and then test in several careful experiments to better understand the algorithm.** If we took the whole system and showed that it performed amazingly well from scratch in Atari or Mujoco, well that is justification in itself that the idea is useful and should be pursued. However, when proposing a new approach, necessarily we often won’t yet have all the algorithms we need, and thus we cannot demonstrate that the whole system is better than the current SOTA. All we can do is motivate why it could be better, hand-specify or engineer parts of the system for which we do not yet have great solutions (but should be able to develop them) and show that if we had such algorithms we would get the benefits we claim the framework provides. **An important step is to understand potential of this new planning framework and its characteristics to guide future algorithmic development for it.** If we know we need to learn many options and value functions off-policy for subgoals, then we can improve such algorithms. If we know we need representations that can give us different levels of granularity for planning, then we can focus on developing algorithms for these things.

These focused experiments we perform may all be in Pinball, but we view such testing not as a limitation. The Pinball environment allow us to vary many conditions and better visualize the behavior of the algorithm to develop deeper insights. **We view this careful testing as a starting point for further empirical studies and algorithmic development.**

We summarized the many experiments we ran to better understand GSP, in the conclusion:
“We show, in the PinBall environment, that 1) the subgoal-conditioned models can be accurately learned using standard value estimation algorithms (Section 4.3), 2) GSP can significantly improve speed of learning over DDQN (Figure 5) and 3) outperforms several Dyna variants, including Dyna with options (Appendix G). Additionally, 4) GSP can still benefit from less accurate models (Figure 8) and 5) relearns more quickly under nonstationarity than DDQN (Figure 9).”
**These results try to show evidence for our claims (learnability of models, robustness to errors in models, learning speed improvements, etc) while providing us the insights that we need to further develop this framework.**

We thank you for taking the time to read our paper and this response. We will of course use your suggestions to improve the paper.

---

### Decision · Program_Chairs · 2023-01-20

**Decision:**

Reject

**Justification For Why Not Higher Score:**

Reviewers raised a number of concerns about the novelty, results, and technique details. The authors fail to address these issues.

**Justification For Why Not Lower Score:**

N/A

**Metareview: Summary, Strengths And Weaknesses:**

Summary:
The paper proposes "goal-space planning" to improve the efficiency of RL. This approach takes a number of sub-goals as abstract states and trains a model-free policy that uses user-specified subgoal states to perform approximate longer-horizon bellman updates and thus accelerate learning. The paper argues that this approach side-steps many problems in traditional model-based planning approaches, such as the inaccuracy of the learned model.

Strength:
- The paper proposes an interesting approach to incorporate subgoal value estimates into Q-learning updates.
- The paper was clearly written and it is easy to follow.

Weakness:
- Reviewers worried that the proposed approach is not novel since similar ideas have been seen in the literature
- Assumptions of known subgoals seem to be too strong
- Results might not be significant enough